# Investigating The Functional Roles of Attention Heads in Vision Language Models: Evidence for Reasoning Modules

## Abstract

Despite excelling on multimodal benchmarks, vision–language models (VLMs) largely remain a black box. In this paper, we propose a novel interpretability framework to systematically analyze the internal mechanisms of VLMs, focusing on the functional roles of attention heads in multimodal reasoning. To this end, we introduce CogVision, a dataset that decomposes complex multimodal questions into step-by-step subquestions designed to simulate human reasoning through a chain-of-thought paradigm, with each subquestion associated with specific receptive or cognitive functions such as high-level visual reception and inference. Using a probing-based methodology, we identify attention heads that specialize in these functions and characterize them as functional heads. Our analysis across diverse VLM families reveals that these functional heads are universally sparse, vary in number and distribution across functions, and mediate interactions and hierarchical organization. Furthermore, intervention experiments demonstrate their critical role in multimodal reasoning: removing functional heads leads to performance degradation, while emphasizing them enhances accuracy. These findings provide new insights into the cognitive organization of VLMs and suggest promising directions for designing models with more human-aligned perceptual and reasoning abilities. Code is at https://anonymous.4open.science/r/ICLR-C327.

## 1 Introduction

Large Vision-Language Models (VLMs) (Zhu et al., 2023; Liu et al., 2023; Lu et al., 2024a) have demonstrated remarkable success across diverse multimodal tasks, ranging from image captioning to visual question answering. Although VLMs can solve mathematical reasoning problems with visual context (as shown in Figure 1), their internal mechanisms remain poorly understood.

For humans, solving such complex problems (illustrated in Figure 1) typically requires the collaboration of vision and language, engaging multiple brain regions (Barsalou, 2014): the occipital lobe for visual reception, capturing and processing the content of the images; the temporal lobe supports long-term memory and the recall of relevant factual knowledge, such as chemical concentration formulas (Wheeler et al., 1997); and the parietal and prefrontal cortices are involved in higher-order reasoning (Hubbard et al., 2005), to produce the correct answer.

Recent research in interpretability has begun probing the internal organization of large language models (LLMs), revealing specialized attention heads for specific functions (Wu et al.; Li et al., 2023a; Zheng et al.). In the case of VLMs, several studies (Kang et al., 2025; Bi et al., 2025) have identified sparse attention heads with special functional roles in tasks such as grounding. However, studying VLMs in complex, multi-step reasoning scenarios remains underexplored. A deeper understanding of whether such specialized components exist, how they are organized, and what functional roles they play in multimodal reasoning is therefore critical.

In this paper, we propose a novel interpretability framework for systematically analyzing the functional roles of attention heads-parallel units in transformer models that compute token-to-token attention-an important component in VLMs, with a focus on their contributions to reception (perceptual processing) and cognition. To facilitate this, we introduce CogVision, a dataset that bridges the gap between model analysis and human cognitive processes. CogVision decomposes multimodal

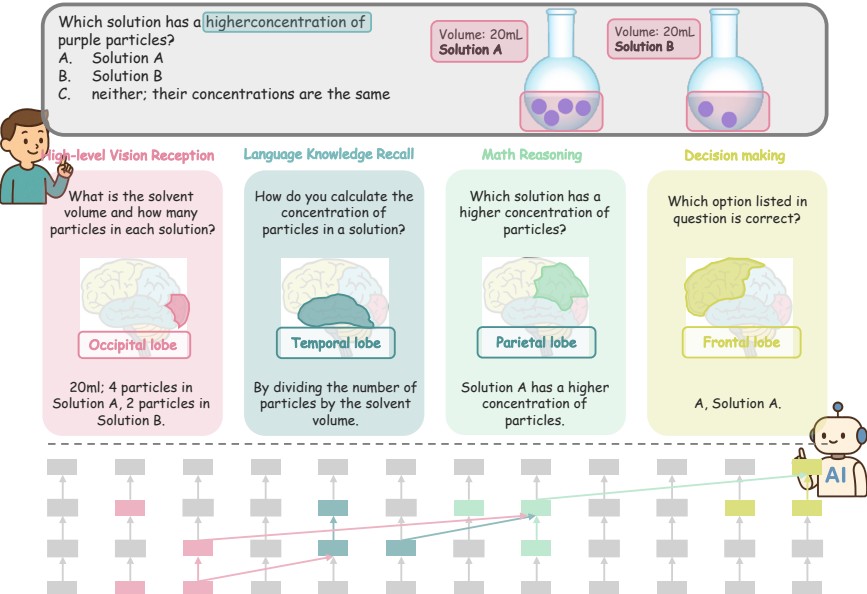

Figure 1: To answer a complex question, the human brain engages multiple regions, each performing distinct cognitive functions. We investigate whether specific attention heads in large vision language models play analogous functional roles in generating responses.

queries into step-by-step subquestions, each aligned with specific cognitive functions (such as math reasoning, decision-masking), thus enabling a fine-grained evaluation of reasoning aligned with the chain-of-thought (CoT) paradigm. Leveraging CogVision, we develop a probing method to identify and characterize attention heads responsible for distinct cognitive operations across vision and language within the transformer architecture.

We conduct extensive experiments on three major VLM families, including Intern (Zhu et al., 2025), Qwen (Yang et al., 2025), and Gemma (Team et al., 2025) with different model scales. Our results reveal the existence of cognitive heads that consistently exhibit **universal**, **sparse**, and **intrinsic** properties across architectures. Further analysis of the correlations among these functional heads reveals **cross-function interactions**, where a single head may support multiple functions or modalities, and uncovers a **hierarchical structure** in which lower-level functional heads modulate higher-level ones, showing the complexity of neural networks (Barsalou, 2014; Ono et al., 2022).

Furthermore, we validate the functional importance of these heads by showing that their removal degrades performance on complex tasks and leads to specific error patterns, while their enhancement improves reasoning capabilities. Our findings provide compelling evidence that these attention heads play a critical role in multimodal reasoning. This insight not only deepens our understanding of the internal organization of VLMs but also suggests potential avenues for designing more interpretable and cognitive-inspired multimodal AI systems.

## 2 COGVISION

In this section, we present our dataset Cognitive Vision (CogVision) that contains cognitive process in multimodal reasoning. CogVision contains 1,409 main questions and 5,744 subquestions. Each example comprises a main question and answer, its subquestions and subanswers and an annotation specifying the receptive or cognitive function required for each subquestion.

### 2.1 COGNITIVE FUNCTIONS

To systematically capture the cognitive processes involved in complex reasoning tasks, we consider eight functions related to complex multimodal reasoning. These functions are inspired by

established frameworks in cognitive science (Anderson, 2014; Diamond, 2013), which highlight the importance of perception, working memory, and reasoning in human cognition.

The functions related to vision include:

- **Low-level Visual Reception**: Recognizing basic visual features such as color, shape, size, position, motion.
- **High-level Visual Reception**: Integrating visual information to recognize objects, patterns, and scene structure.
- **Visual Knowledge Recall**: Applying long-term visual knowledge related to visual concepts and their properties.

The functions related to language include:

- **Language Information Extraction and Understanding**: locating and understanding relevant information from an external source or prior context.
- **Language Knowledge Recall**: Accessing domain-specific knowledge without visual input.
- **Math Reasoning**: Performing counting, arithmetic, comparison, and logic-based operations.
- **Inference**: deriving implicit information that is not directly stated.
- **Decision-Making**: Selecting the best outcome among alternatives based on reasoning.

This categorization reflects a natural progression from basic information processing to complex cognitive integration. Both the human brain and VLMs encompass a wide range of functional modules. Our focus in this work is specifically on reasoning-related cognitive functions. By identifying and organizing these eight core reasoning functions, we can more clearly examine how VLMs handle different types of thinking steps, in a way that is both systematic and easy to interpret. Detail descriptions of each function and examples in CogVision can be found in Appendix A.2.

## 2.2 DATA COLLECTIONS

Based on our categorization of reasoning functions, we sampled 2000 diverse questions from existing commonly used visual reasoning benchmarks, selecting 200 examples each from Clevr-Math (Lindström & Abraham, 2022), MathVision (Wang et al., 2024a), MathVista (Lu et al., 2024b), MMMU (Yue et al., 2024), ScienceQA (Saikh et al., 2022), OKVQA (Marino et al., 2019) (400 examples), VCR (Zellers et al., 2019), VisuLogic (Xu et al., 2025) and Super-Clevr (Li et al., 2023b) datasets. These datasets span a range of reasoning problems, including logical, mathematical, and commonsense reasoning. Using the chain-of-thought (CoT) paradigm, we prompted GPT-4.1 (Achiam et al., 2023) to decompose each main question (with its final answer) into subquestions, each targeting a subanswer and a specific function. The prompt encourages structured, step-by-step reasoning, ensuring that each subquestion is clear, answerable, and sequentially dependent. This process yields a set of subquestion–answer–function (subQAF) triples for each QA pair: $\text{subQAFs} = \{(q_i, a_i, f_i)\}_{i=1}^{k}$, where each contains a subquestion $q_i$, its concise answer $a_i$, and the corresponding function label $f_i$. The prompt for generating subquestions are list in Appendix A.7.

## 2.3 DATA FILTERING AND ANNOTATION

Recent advances enable leveraging large pre-trained models for dataset construction, thanks to their reasoning capabilities and ability to generate high-quality annotations at scale (Wang et al., 2024b). While our dataset is automatically generated using a VLM to minimize manual effort, we employ a rigorous two-stage human verification pipeline to ensure data quality and mitigate hallucination. In the first stage, three expert annotator independently evaluate whether each subquestion is logically structured and consistent with natural human reasoning. QA pairs with incoherent or inconsistent decompositions are moved out. In the second stage, annotators verify and, if necessary, relabel the cognitive function associated with each subquestion to ensure accurate alignment with the intended mental process. Subanswers are further cross-checked with GPT-o3 (OpenAI, 2024) and adjudicated by humans in cases of disagreement (details in Appendix A.6). This multi-step verification ensures that each retained subQAF triplet represents a coherent, interpretable reasoning step grounded in core cognitive functions. After this refinement, our final dataset comprises 1,409 main QA pairs and 5,744 validated subQAF triplets. The detailed statistics of the dataset including the number of

triplet in training and testing set and distributions of each function can be found in Appendix A.4. We analyze in-group subquestion diversity, and the results in the Appendix A.3 show that the eight functions exhibit broad and overlapping distributions in phrasing patterns and token lengths, indicating no systematic surface-form differences.

# 3 DETECTION OF COGNITIVE FUNCTIONS

Using the CogVision dataset, we adopt a probing-based framework (Alain & Bengio, 2016; Belinkov, 2022; Tenney et al., 2019) to identify which attention heads in VLMs are associated with specific functions in reasoning process. Specifically, for each functional annotated subquestion, we extract head activations (see Subsection 3.1), train classifiers and compute accuracies to identify contributing heads (see Subsection 3.2). Unlike prior work focusing on a single label, our formulation captures many-to-many relationships between heads and cognitive functions, enabling a more nuanced analysis of functional specialization and overlap within the model.

## 3.1 HEAD FEATURE EXTRACTION

Given a large VLM $\mathcal{M}$, we generate an answer $a_i^{\mathcal{M}}$ for each subquestion $q_i$ derived from a main question $Q_i$. To support coherent multi-step reasoning, we include preceding subquestions and their answers as contextual input, emulating the incremental reasoning process observed in human cognition.

During inference, each input token is first mapped into an embedding and then propagated through the transformer's layers. At each layer, attention and feedforward operations update the residual stream, which is ultimately decoded into token predictions. For each generated token $i$, we extract attention head outputs $X_i = \{x_l^m \mid l \in 1, \ldots, N_l, \ m \in 1, \ldots, N_h\}$ across all layers, where $x_l^m$ denotes the value vector from the $m$-th head in layer $l$ projected into the residual stream, with $N_h$ the number of heads per layer and $N_l$ the total number of layers.

Let $N_t$ denote the number of tokens in the generated answer $a_i^{\mathcal{M}}$. To isolate semantically informative content relevant to reasoning, we select the top-$k$ most important tokens, determined by prompting Qwen3-30B LLM (Yang et al., 2025), yielding an index set $\mathcal{I}_k$ with $|\mathcal{I}_k| = k$. For each index $j \in \mathcal{I}_k$, we extract the corresponding attention head activations $X_j$, and compute the averaged activation feature for the $m$-th head in layer $l$ as $\bar{x}_l^m = \frac{1}{k} \sum j \in \mathcal{I}_k x_l^m$. This results in a full set of head-level features $\bar{X}_i = \{\bar{x}_l^m \mid l \in 1, \ldots, L, \ m \in 1, \ldots, M\}$.

## 3.2 FUNCTION PROBING

For the dataset with $N$ subQAF triplets, we collect all activations to construct the probing dataset:

$$\mathcal{D}_{\text{probe}} = \{(\bar{x}_l^m, \ c)_i\}_{i=1}^N, l \in \{1, \ldots, L\}, \ m \in \{1, \ldots, M\} \tag{1}$$

For classification based on CogVision, the training set includes 1,124 main questions with 4,604 subQAF triplet, while the testing set has 285 main questions with 1,141 triplets. Our probe takes the form $p_\theta(x_l^m) = \text{sigmoid}(\langle \theta, x_l^m \rangle)$. There is one probe per attention head per layer per function. For each target function, the probe is trained by treating the attention-head outputs that lead to correct answers for that function as the positive class, and those associated with correct answers from other functions as the negative class. To ensure data balance, we select an equal number of negative samples to match the positive ones. Given prior findings suggesting that cognitive functions may vary by layer depth (Zheng et al.), we incorporate layer-wise information by computing the average activation $\bar{x}_l = \frac{1}{M} \sum_{m=1}^M \bar{x}_l^m$ for each layer. We then augment each head-level vector with its corresponding layer summary, resulting in enriched features $x_l^{m'} = [\bar{x}_l^m; \bar{x}_l]$ for probing. The importance for each head are then calculated based on the accuracy of predicting target function. The effectiveness of top-k tokens and layer information, as well as the sensitivity analysis with respect to the parameter $k$ and the choice of LLM fused for top-k token extraction, and prompt format, can be found in Appendix A.9.

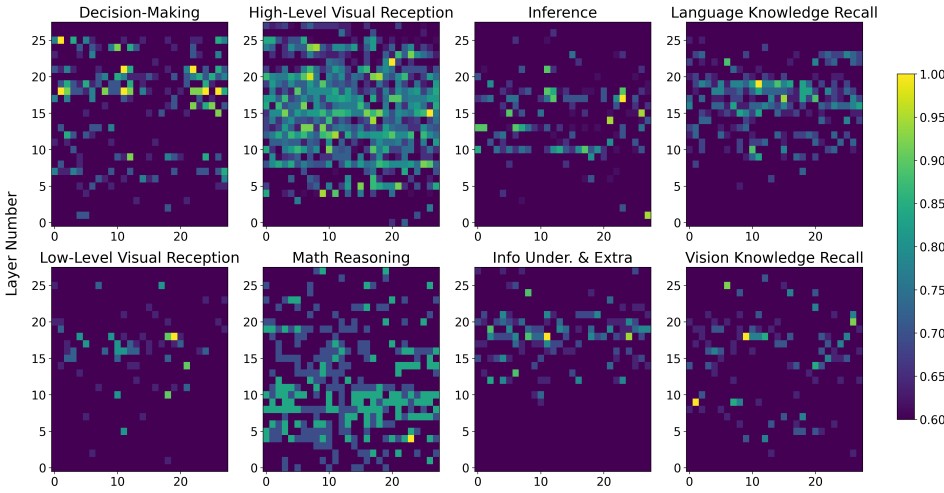

Figure 2: The existence of cognitive heads in Qwen2.5-VL-7B responsible for eight distinct functions in complex reasoning tasks. The x-axis represents the head index, while the y-axis indicates the layer index. The values denote head importance scores, capped at a cutoff of 0.60.

## 4 EXPERIMENTS

We conduct a series of experiments on three VLM families across various model scales, including Intern (Zhu et al., 2025) (InternVL3-8B and InternVL3-2B), Qwen (Yang et al., 2025) (Qwen2.5-VL-7B and Qwen2.5-VL-3B), and Gemma (Team et al., 2025) (Gemma3-4B and Gemma3-2B). We analyze the commonalities and differences of functional heads (Subsection 4.1), validate their contributions (Subsection 4.2), and examine correlations, including cross-function interactions and hierarchical organization (Subsection 4.3). We also assess their causal impact on downstream reasoning tasks (Subsection 4.4). Results confirm the existence of sparse, function-specific heads and highlight their critical contribution to structured cognitive processing within VLMs.

### 4.1 PROPERTIES OF COGNITIVE HEADS

**Sparsity, Universality, and Intrinsic Organization:** Figure 2 shows the heatmap of attention head accuracy across eight functions in Qwen2.5-VL-7B on the CogVision test set, revealing a sparse distribution. In total, fewer than 7% of all heads achieve accuracies above 0.9 across the eight functions (about 2% for high-level visual reception and math reasoning, and less than 1% for the others), suggesting that only a small subset of heads meaningfully contributes to different reasoning tasks. These results demonstrate that VLMs rely on highly specialized, localized components for distinct cognitive abilities. Pearson correlations between head-activation heatmaps across the eight functions (Figure 3) are generally low, confirming that different functions tend to depend on partially separable subsets of heads. Moreover, this sparse functional organization is consistent across architectures and scales: heatmaps for five additional models (Appendix A.5) confirm its universality, and the relatively high Pearson correlation coefficients between models further verify this consistency (in Appendix A.10). Within the same model family (e.g., Qwen2.5-VL-7B in Figure 2 vs. Qwen2.5-VL-3B in Figure 8), we observe similar distributions, suggesting that such specialization is intrinsic to VLMs.

**Functional Personalization:** Beyond sparsity, attention heads exhibit a structured distribution across model layers. Math-related heads are dispersed throughout the network, whereas inference-related heads appear more frequently in higher layers. This task-dependent localization suggests an emergent modular organization in which different layers support distinct cognitive operations. We also observe notable variation in head counts across functions. For example, in the Qwen family, math reasoning and high-level visual reception heads are more prevalent than others, reflecting differences in representational and computational complexity. Smaller models contain fewer functional heads compared to their larger counterparts.

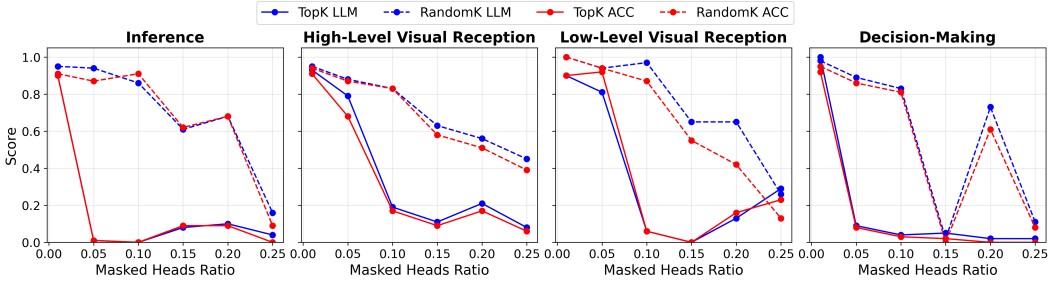

Figure 3: Pearson Correlation between different functions across two models.

Figure 4: The performance of Qwen2.5-VL-3B after masking out top K cognitive heads vs K random heads on inference, high-level visual reception, low-level visual reception, and decion-making.

## 4.2 Functional Contributions of Cognitive Heads

After identifying the cognitive heads associated with each function, we examine their functional roles by evaluating the model's behavior on the CogVision test set under targeted interventions. We perform head ablation by scaling the output of a specific attention head with a small factor $\epsilon$ (e.g., 0.001), effectively suppressing its contribution:

$$x_i^{\mathrm{mask}} = \mathrm{Softmax}\left(\frac{W_q^i W_k^{iT}}{\sqrt{d_k/n}}\right) \cdot \epsilon W_v^i \qquad (2)$$

Specifically, we compare model performance when masking identified cognitive heads versus masking an equal number of randomly-selected heads. To quantify the impact, we employ both an LLM-based judge and an integrated accuracy metric. For LLM-based judge, we use LLM (Qwen3-30B LLM (Yang et al., 2025)) to judge the correctness of the output. For the integrated accuracy metric, an output is considered unaffected if its BLEU score (Papineni et al., 2002) exceeds 0.8, or if either the ROUGE score (Chin-Yew, 2004) or the semantic similarity score surpasses 0.6. This provides a comprehensive evaluation of performance degradation.

We gradually mask out the number of cognitive heads and and observe how model behavior changes. As shown in Figure 4, randomly masking up to 10% of heads has minimal impact on the overall performance of Qwen2.5-VL-3B. In contrast, masking a similar number of cognitive heads leads to a substantial drop across multiple functions. Notably, when more than 25% of heads are randomly masked, performance also declines sharply, as this begins to include functionally critical heads. These results further highlight the sparsity and importance of functional heads.

For each function, we select the top 10% of heads with the highest accuracy as cognitive heads. As shown in Table 1, masking cognitive heads lead to a substantial decline in performance, whereas masking an equal number of random heads results in only minor degradation across all VLMs. In some cases, masking identified cognitive heads reduces accuracy to zero, indicating that the model cannot perform the corresponding function without them. The t-test analysis (Appendix A.10) shows that the difference between cognitive masking and random masking is statistically significant, with

Table 1: Intervention results (mean accuracy over 5 runs %) of cognitive heads vs. random heads across 8 functions: **Low-level** Visual Reception, **High-level** Visual Reception, Vision Knowledge **Recall**, Language **Info**rmation Extraction and Understanding, Language Knowledge **Recall**, **Math** Reasoning, **Inference**, and **Decision** Making. For model names, Intern2B: InternVL3-2B, Intern8B: InternVL3-8B, gemma2B: gemma-3n-e2b-it, gemma4B: gemma-3n-e4b-it, Qwen3B: Qwen2.5-VL-3B-Instruct, Qwen7B: Qwen2.5-VL-7B-Instruct. Lower values indicate more effective intervention outcomes, suggesting that the corresponding heads play a greater role in the cognitive function.

| Model | Inter_Head | Vision mainly Functions | | | | | | Language mainly Functions | | | | | | | | | |
| | | Low-Level | | High-Level | | Recall | | Info | | Recall | | Math | | Inference | | Decision | |
| | | llm | acc | llm | acc | llm | acc | llm | acc | llm | acc | llm | acc | llm | acc | llm | acc |
| Intern2B | random | 75.61 | 82.44 | 87.5 | 88.75 | 89.15 | 86.1 | 57.84 | 69.19 | 84.06 | 84.64 | 81.29 | 90.97 | 75.06 | 74.12 | 67.22 | 71.67 |
| | cognitive | 60.24 | 62.68 | **75.71** | **76.61** | **73.05** | **64.58** | 66.76 | 68.92 | **44.93** | **46.38** | **61.29** | **64.52** | **71.76** | **64.71** | **48.06** | **52.22** |
| Intern8B | random | 92.2 | 93.17 | 88.47 | 91.25 | 87.61 | 82.25 | 59.41 | 69.12 | 87.59 | 89.37 | 79.25 | 84.15 | 82.64 | 85.05 | 66.27 | 73.49 |
| | cognitive | **68.78** | **78.05** | **56.94** | **65.97** | **71.69** | **70.56** | **8.82** | **19.12** | **74.68** | **77.22** | **20.75** | **56.6** | **43.96** | **42.86** | 66.27 | **66.27** |
| gemma2B | random | 58.37 | 76.33 | 57.53 | 67.64 | 54.57 | 62.57 | 55.07 | 60.82 | 81.05 | 82.11 | 23.12 | 55.0 | 57.44 | 63.49 | 30.0 | 66.94 |
| | cognitive | **48.98** | **55.1** | **55.06** | **63.48** | **2.86** | **8.57** | **30.27** | **38.49** | **50.0** | **47.37** | **11.25** | **36.88** | **36.98** | **52.09** | **19.44** | **54.17** |
| gemma4B | random | 36.0 | 48.73 | 29.22 | 38.65 | 33.52 | 34.08 | 25.12 | 27.56 | 40.27 | 47.84 | 27.91 | 44.19 | 57.3 | 57.08 | 18.29 | 34.57 |
| | cognitive | **29.09** | **41.82** | **10.88** | **21.24** | **5.63** | **14.08** | **22.2** | **36.34** | **9.46** | **13.51** | 27.91 | **53.49** | **36.18** | **32.81** | **4.29** | **15.71** |
| Qwen3B | random | 70.97 | 82.58 | 82.42 | 86.06 | 88.48 | 86.36 | 52.22 | 58.89 | 85.14 | 86.57 | 65.85 | 89.76 | 85.32 | 90.13 | 63.44 | 71.88 |
| | cognitive | **12.9** | **12.9** | **12.12** | **16.67** | **77.88** | **77.88** | 55.56 | 62.96 | **61.43** | **68.57** | **0.0** | **0.0** | **1.27** | **1.27** | **1.56** | **4.69** |
| Qwen7B | random | 83.2 | 88.8 | 84.57 | 89.51 | 79.43 | 80.29 | 75.08 | 79.38 | 90.13 | 86.4 | 67.84 | 72.94 | 80.67 | 83.33 | 75.14 | 79.19 |
| | cognitive | **30.0** | **38.0** | **73.21** | **73.83** | **21.43** | **22.86** | **15.38** | **33.85** | **84.0** | **78.67** | 68.63 | 72.55 | 85.56 | 81.11 | **25.68** | **27.03** |

$p \ll 0.05$ in nearly all cases. To further validate their functional roles, we mask heads associated with one function (e.g., language knowledge recall) while evaluating performance on a different function (e.g., vision knowledge recall). As shown in Figure 5, masking the relevant functional heads yields a significantly larger performance drop than masking unrelated heads, confirming their functional specialization.

In addition to masking, we also conduct activation patching, where the activations of cognitive heads associated with one function are replaced by those from another function using two strategies: random activation and mean activation. In the random activation setting, activations are substituted with those from a randomly selected subquestion belonging to a different function. In the mean activation setting, activations are replaced with the average activation computed over all subquestions associated with another function (details in Appendix A.11). As shown in Table 2, both types of activation patching result in substantial performance degradation for cognitive heads, consistent with the effects observed under masking interventions.

Table 2: Ablation study of different activation-masking methods on Qwen2.5-VL-3B. Random: random activation. Mean: mean activation. Scalar: masking.

| Method | Inter_Head | Vision mainly Cognitive Functions | | | | | | Language mainly Cognitive Functions | | | | | | | | | |
| | | Low-Level | | High-Level | | Recall | | Info | | Recall | | Math | | Inference | | Decision | |
| | | llm | acc | llm | acc | llm | acc | llm | acc | llm | acc | llm | acc | llm | acc | llm | acc |
| Random | cognitive | **0.00** | **0.00** | 20.45 | 23.48 | **62.12** | **65.15** | 35.79 | 57.41 | 12.86 | 24.29 | 17.07 | 17.07 | 14.32 | 14.32 | 6.25 | 9.38 |
| Mean | cognitive | 3.80 | 3.80 | 17.39 | 19.91 | 66.67 | 65.15 | **35.19** | **55.56** | **10.00** | **21.43** | 37.17 | 34.15 | 3.80 | 3.80 | 6.25 | 9.38 |
| Scalar | cognitive | 6.45 | 6.45 | **16.67** | **18.94** | 75.76 | 75.76 | 62.96 | 81.48 | 57.14 | 62.86 | **2.43** | **2.43** | **0.00** | **0.00** | **3.13** | **4.69** |

We further examine how vision-related and language-related heads attend to their respective modalities. For the top-30 heads of each function, we compute the average attention weight on visual tokens across the test set. As shown in Appendix A.10, vision-related heads (e.g., high-level and low-level visual reception) predominantly focus on image tokens, capturing spatial and object-level information, whereas language-related heads (e.g., language knowledge recall) concentrate on text tokens. Interestingly, Qwen and Intern models allocate more attention to text, while Gemma emphasizes vision, revealing family-specific modality preferences. We also observe heads with cross-modal attention that respond to both visual and textual tokens, likely mediating interactions between visual

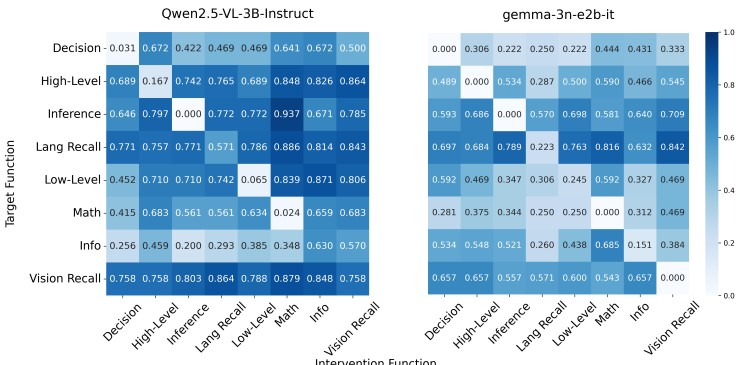

Figure 5: The performance of VLMs on target functions after masking out top K cognitive heads on intervention functions. The scores are based on LLM-Judge.

Table 3: Study on the influence of low-level cognitive heads for high-order function on Qwen2.5-VL-3B. The score is measured based on LLM-judge. We only evaluate subquestions that the model originally answered correctly. This filtering ensures that any observed drop in performance is caused solely by the intervention. Notably, the model's own generated outputs are used as inputs for subsequent subquestions.

| Vision Recall | Language Recall | Info. | Low-Level | High Level | Math | Decision | Inference |
|---|---|---|---|---|---|---|---|
| ✗ | ✓ | ✓ | ✓ | ✓ | 50.00 ↓ 50.00 | 54.55 ↓ 45.45 | 54.17 ↓ 45.83 |
| ✓ | ✗ | ✓ | ✓ | ✓ | 16.67 ↓ 83.33 | 56.25 ↓ 43.75 | 65.22 ↓ 34.78 |
| ✓ | ✓ | ✗ | ✓ | ✓ | 22.22 ↓ 77.78 | 57.89 ↓ 42.11 | 51.61 ↓ 48.39 |
| ✓ | ✓ | ✓ | ✗ | ✓ | 27.27 ↓ 72.73 | 72.73 ↓ 27.27 | 59.09 ↓ 40.91 |
| ✓ | ✓ | ✓ | ✓ | ✗ | 33.96 ↓ 66.04 | 64.29 ↓ 35.71 | 53.95 ↓ 46.05 |

perception and linguistic reasoning. These findings suggest that functional specialization in VLMs is complemented by selective cross-modal integration, enabling coherent multimodal reasoning.

## 4.3 RELATIONSHIP AMONG COGNITIVE HEADS

While cognitive heads are specialized for distinct functions, understanding their relationships is crucial for revealing how complex reasoning emerges from their cooperation.

**Heads Across Functions** The neural system is inherently complex, with individual neurons often participating in multiple functions (Mante et al., 2013). We observe a similar phenomenon in VLMs: certain functional heads overlap, with a single head participating in multiple cognitive roles (e.g., in the Qwen2.5-VL-7B model, 18% of cognitive heads across eight functions participate in more than one function). In our probing-based method, we quantify and rank the accuracy of attention heads for each cognitive function. A head that ranks highly for one function may also exhibit non-negligible importance for others, leading to the phenomenon of "Heads Across Functions". Notably, even if a head ranks in the top 10% for multiple cognitive functions, our ranking still reveals a primary function for which it is most diagnostic. In summary, functional heads serve not only as specialized units but also as integrative components that bridge multiple reasoning processes.

**Hierarchical Structure** Humans often solve complex tasks through step-by-step reasoning, where former functions, such as low-level visual reception, support higher-level processes like inference and decision making. The CogVision dataset reflects this hierarchy: under CoT, early subquestions focus on information extraction, while later ones require more complex reasoning. Leveraging this structure, we test whether VLMs exhibit similar functional dependencies by masking attention heads associated with early-stage functions and observing the impact on subsequent reasoning steps. As shown in Table 3, masking vision or language knowledge recall heads significantly impairs later-stage performance, particularly in decision making. These results suggest that VLMs exhibit an

Table 5: Positive Intervention on CogVision test set and other Visual Question Answering benchmarks (OK-VQA, MathVista and Visulogic). The scores are based on LLM-Judge. For OK-VQA, we perform positive intervention by masking 10% high-level visual reception head, MathVista for math reasoning heads, and Visulogic for decision-making heads.

| Model | Inter_Head | In Domain | | | | | | | | Out of Domain | | |
|---|---|---|---|---|---|---|---|---|---|---|---|---|
| | | Math | Vision Recall | Lang Recall | Info | Low-Level | Inference | High-Level | Decision | OK-VQA | MathVista | Visulogic |
| Qwen3B | before | 46.40 | 82.29 | 81.37 | 48.03 | 78.26 | 65.87 | 77.38 | 29.73 | 62.50 | 60.00 | 26.50 |
| | after | 46.40 | **85.42** | **84.31** | 44.74 | 78.26 | **69.84** | **77.98** | **32.43** | 62.00 | 60.00 | **28.00** |
| Qwen7B | before | 52.00 | 84.38 | 86.27 | 43.42 | 77.18 | 69.84 | 82.44 | 36.94 | 66.00 | 63.00 | 24.00 |
| | after | **52.80** | **85.42** | 86.27 | **46.05** | **82.61** | **74.60** | 82.44 | 36.94 | **67.50** | 63.00 | **24.50** |
| Intern2B | before | 38.40 | 79.17 | 80.39 | 44.08 | 78.26 | 61.90 | 81.55 | 34.23 | 58.50 | 51.50 | 24.00 |
| | after | 38.40 | 78.13 | **84.31** | 42.76 | **80.43** | **64.29** | **82.44** | **35.14** | **61.50** | **52.00** | **26.00** |
| Intern8B | before | 52.00 | 88.54 | 82.35 | 45.39 | 88.04 | 73.81 | 86.61 | 42.34 | 67.00 | 66.00 | 24.00 |
| | after | **52.80** | **91.67** | **84.31** | **46.71** | 86.96 | 73.81 | **87.80** | **43.24** | **67.50** | 66.00 | **26.00** |
| gemma2B | before | 26.40 | 62.50 | 80.39 | 32.24 | 34.78 | 48.41 | 33.93 | 23.42 | 29.00 | 24.50 | 26.00 |
| | after | **28.00** | **64.58** | **84.31** | **38.16** | 34.78 | **50.00** | 30.95 | 20.72 | **29.50** | 24.50 | **29.00** |
| gemma4B | before | 32.80 | 62.50 | 83.33 | 32.89 | 27.17 | 52.38 | 33.63 | 18.92 | 31.50 | 24.00 | 27.00 |
| | after | **35.20** | **65.63** | **85.29** | **35.53** | 27.17 | **53.97** | 33.63 | **19.82** | **33.00** | **26.50** | **27.50** |

emergent hierarchical organization, where early cognitive functions support more advanced reasoning. The prompt used for VLMs can be found in Appendix A.8.

## 4.4 INFLUENCE OF FUNCTIONAL HEADS ON DOWNSTREAM TASKS

In this section, we investigate how functional heads influence downstream tasks through both negative interventions (masking out function heads) and positive interventions (shifting heads toward specific functions).

**Negative Intervention:** We randomly sample 200 question for two VQA benchmarks, OK-VQA and Clevr, both reasoning tasks. We perform negative intervention by masking high-level visual reception heads on OK-VQA and math reasoning heads on Clevr-Math, effectively suppressing their activations. As shown in Table 4, masking these key cognitive heads leads to a significant performance drop across all models. Further analysis in Ap-

Table 4: Negative Intervention on Visual Question Answering task (OK-VQA and Clevr-Math). The scores are based on LLM-Judge.

| Dataset | Inter_Head | Model | | | | | |
|---|---|---|---|---|---|---|---|
| | | Qwen3B | Qwen7B | Intern2B | Intern8B | gemma2B | gemma4B |
| OK-VQA | before | 54.00 | 55.00 | 46.00 | 51.00 | 21.00 | 20.00 |
| | after | **7.00** | 55.00 | **44.00** | 53.00 | **18.00** | 21.00 |
| Clevr-Math | before | 94.00 | 70.00 | 20.00 | 93.00 | 14.00 | 14.00 |
| | after | **0.00** | **59.00** | 29.00 | **13.00** | 14.00 | **13.00** |

pendix A.12 shows that masking the **math reasoning** heads leads to errors in arithmetic tasks, while visual receptive functions remain largely unaffected. This confirms that these cognitive heads are crucial for specific functions and highlights the robustness and generalizability of our method.

**Positive Intervention:** We calculate the activation directions of different functions using the CogVision dataset. For each function, the activation direction of a head at layer $l$ and index $h$ is computed as:

$$\text{dir}_l^h = \mathbb{E}_{i \in \mathcal{D}_{\text{correct}}} \left[ x_l^h(i) \right] - \mathbb{E}_{i \in \mathcal{D}_{\text{incorrect}}} \left[ x_l^h(i) \right] \tag{3}$$

where $x_l^h(i)$ denotes the activation of head at layer $l$ and index $h$, and $\mathcal{D}_{\text{correct}}$ and $\mathcal{D}_{\text{incorrect}}$ represent the sets of samples answered correctly and incorrectly, respectively. Then we estimate the standard deviation of activations (Li et al., 2023a) along the cognitive function direction to be $\sigma_l^h$, and shift original head activation as $x_l^h(i) \leftarrow x_l^h(i) + \alpha \sigma_l^h \text{dir}_l^h$, where $\alpha$ is a parameter.

The experimental results in Table 5 show that enhancing the activation of functional heads along their corresponding functional directions improves performance on the related tasks. For example, positive intervention on vision knowledge recall heads in InternVL3-8B increased accuracy on the corresponding CogVision question-answering task from 88.54% to 91.67%. Similarly, enhancing function-specific heads can also boost performance on downstream tasks. Here, we set $\alpha = 0.1$ for all datasets, though tuning this parameter may further improve performance. Case analyses are provided in Appendix A.12.

## 5 RELATED WORKS

**Neural Networks and the Brain.** Understanding the relationship between artificial neural networks (ANNs) and the biological brain has been a long-standing goal in both neuroscience and machine learning. Early studies demonstrated that convolutional neural networks (CNNs) trained on visual tasks develop hierarchical representations reminiscent of the ventral visual stream in primates (Yamins et al., 2014; Cadieu et al., 2014). Subsequent work extended this line of inquiry to recurrent and transformer-based architectures, showing that attention mechanisms can emulate aspects of selective processing observed in cortical circuits (Tsividis et al., 2017). More recently, large language models (LLMs) have exhibited striking parallels with human brain activity during language processing. In particular, transformer-based models such as GPT-2 produce internal representations that align with neural responses in language-selective brain regions (Caucheteux et al., 2022; Schrimpf et al., 2021). Some works (Schulze Buschoff et al., 2025; Li et al., 2024) have studied how VLMs perform differently from humans from a cognitive perspective. Furthermore, the chain-of-thought (CoT) paradigm has been argued to mirror step-by-step human reasoning, leading to improved problem-solving performance. These findings motivate the design of interpretable, functionally specialized modules in artificial networks, bridging insights from neuroscience with advances in multimodal reasoning.

**Attention Heads in Vision–Language Models.** A growing body of interpretability research has revealed that attention heads in LLMs exhibit functional specialization, such as pattern induction, truthfulness, information retrieval, and safety alignment (Olsson et al., 2022; Li et al., 2023a; Wu et al.; Zhou et al., 2024; Zheng et al.).

In the multimodal domain, recent works (Li et al., 2020) have begun to explore the internal mechanisms of Vision–Language Models (VLMs). Studies have shown that certain sparse attention heads play distinct roles in visual grounding, enabling alignment between textual tokens and image regions without additional fine-tuning (Kang et al., 2025; Bi et al., 2025). Similarly, probing studies on multimodal pre-trained models (e.g., ViLBERT, LXMERT, UNITER) demonstrate that subsets of attention heads encode cross-modal interactions and semantic alignment between vision and language (Cao et al., 2020). These works highlight the existence of specialized heads in VLMs but largely focus on perception-oriented tasks such as grounding or alignment. In contrast, we investigate functionally specialized heads under more complex reasoning settings by aligning attention head behavior with human cognitive functions.

## 6 CONCLUSION

We propose an interpretability framework that links attention heads in VLMs to human perceptual and cognitive functions involved in multimodal reasoning. To enable this, we introduce CogVision, a cognitively grounded dataset that decomposes complex multimodal questions into functional reasoning steps, and apply probing-based analyses to identify specialized heads supporting these functions. Our study across diverse VLM families reveals that functional heads are sparse, universal, and intrinsic properties of the models, while varying in number, distribution, and hierarchical organization. Moreover, we find that certain heads exhibit cross-modal interactions. Intervention experiments further reveal their causal importance. Our insights into the functional organization of attention mechanisms provide a foundation for developing more interpretable, robust, and cognitively inspired vision-language models. While our work provides a first step toward exploring potential similarities between the cognitive processes of VLMs and those of the human brain, we do not claim complete alignment, nor do we equate observations and analyses of attention heads with the full scope of human reasoning.

**Limitations** While our study provides an initial framework for analyzing attention heads in VLMs, several limitations remain. We focus on eight predefined cognitive functions, which may not cover the full spectrum of LLM capabilities; future work could expand this taxonomy to include finer-grained or emergent functions. Additionally, we concentrate on attention heads, leaving other components such as MLPs unexplored. Further exploring advanced probing methods and extending the analysis to other model components, could provide further understandings.

REPRODUCIBILITY STATEMENT

We provide the CogVision dataset and code at https://anonymous.4open.science/r/ICLR-C327. Upon publication, the repository will be made publicly available.

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

## A APPENDIX

### A.1 THE USE OF LLM

The LLM (GPT-5) was primarily employed for language refinement — including polishing grammar, improving clarity, and rephrasing sentences in the manuscript.

## A.2 COGVISION FUNCTION DETAILS AND EXAMPLES

**Language Information Extraction and Understanding:** The ability to comprehend and extract meaning from only language, including understanding word relationships, sentence structures, context, and intent within a given textual input.

**Low-Level Vision Reception:** The perception and interpretation of visual content, including recognizing low-level visual features such as number, color, shape, size and position.

**High-Level Vision Reception:** The perception and interpretation of visual content, including recognizing high-level visual features such as object recognition, the relationships, motion, spatial arrangement, and scene-level understanding.

**Vision Knowledge Recall:** The access and application of long-term visual knowledge, such as recognizing familiar objects, understanding occlusion, symmetry, physical structure, and part-whole relationships (e.g., "a cat has a tail").

**Language Knowledge Recall:** The access and application of long-term domain-specific textual knowledge, such as factual knowledge from science, history, or everyday concepts stored in memory.

**Math Reasoning:** The application of mathematical concepts and operations such as counting, comparison, arithmetic, and pattern-based quantitative reasoning.

**Inference:** The logical derivation of conclusions from given information, including deductive (guaranteed) reasoning and abductive (plausible) reasoning.

**Decision-Making:** The process of selecting the most appropriate option or answer based on prior reasoning, evaluation of evidence, or predefined objectives.

Table 6 and Table 7 presents illustrative examples from the CogVision dataset. The main question and its corresponding answer are taken from the original dataset. Based on an analysis of the main question, a sequence of sub-questions, their answers, and associated cognitive function labels are generated in order.

## A.3 ANALYSES OF SURFACE-FORM VARIATION ACROSS COGNITIVE-FUNCTION GROUPS

We analyze the surface-form variation across cognitive-function groups. As shown in Figures 6 and 7, the eight functions exhibit wide and overlapping distributions in phrasing patterns and token lengths, indicating no systematic surface-form differences. These results support that the cognitive groups are not determined by trivial lexical or structural artifacts. About modality, as described in Section 2.1, some functions (Low-level Visual Reception, High-level Visual Reception, Visual Knowledge Recall) naturally involve vision, while others relate primarily to language. This modality tendency is intrinsic to the underlying cognitive processes rather than an artifact of the pipeline.

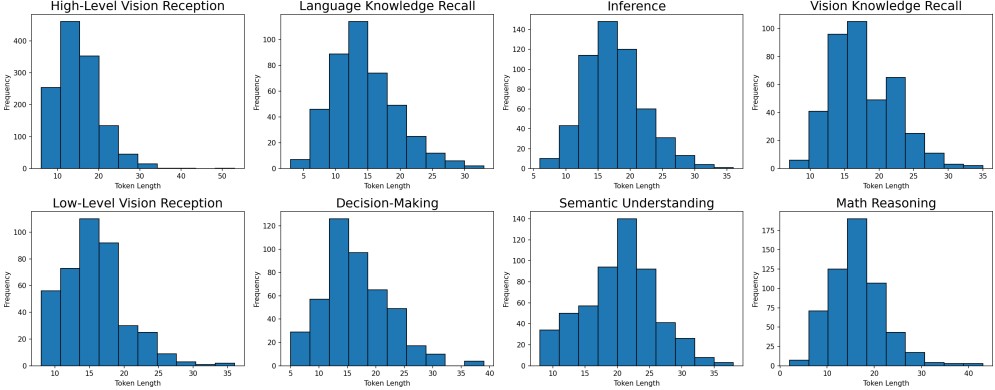

Figure 6: Histogram of token length distribution for 8 functions.

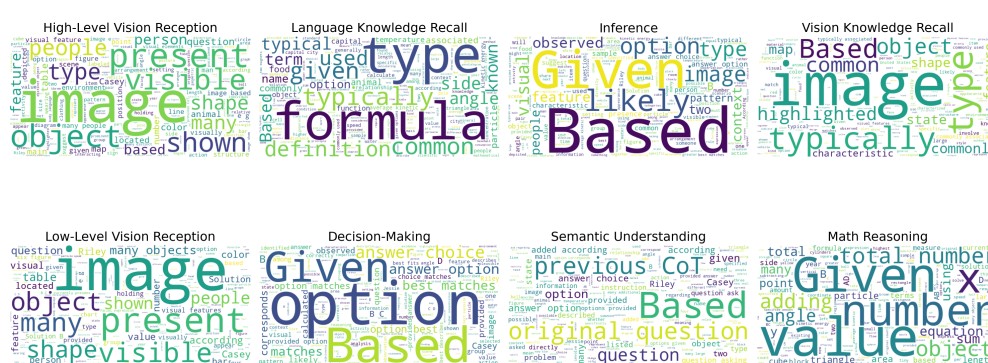

Figure 7: Word cloud distribution for 8 functions.

Table 6: One example from the CogVision dataset showing a main question, its final answer, and a breakdown into subquestions with answers and their corresponding cognitive function labels.

**Example 1:**

| Main Question | In a case-control study, the results were shown in the table below. The OR was: Choose one option from the following: A: 18 B: 16 C: 20D: 10 |
|---|---|
| Answer | B |

| Subquestion | Answer | Cognitive Label |
|---|---|---|
| 1. What are the values in the 2x2 table for cases and controls with and without a history of exposure in the image? | Cases with exposure: 400, Cases without exposure: 100, Controls with exposure: 100, Controls without exposure: 400 | High-Level Vision Reception |
| 2. What is the standard formula to calculate the odds ratio (OR) in a case-control 2x2 table? | OR = (a*d) / (b*c), where a = cases with exposure, b = controls with exposure, c = cases without exposure, d = controls without exposure | Language Knowledge Recall |
| 3. What is the odds ratio (OR) when you substitute the identified values into the formula? | (400*400)/(100*100) = 160,000/10,000 = 16 | Math Reasoning |
| 4. Which option corresponds to the calculated odds ratio? | B: 16 | Decision-Making |

| | Case group | Control group |
|---|---|---|
| A history of exposure | 400 | 100 |
| No history of exposure | 100 | 400 |
| Total | 500 | 500 |

Table 7: One example from the CogQA dataset showing a main question, its final answer, and a breakdown into subquestions with answers and their corresponding cognitive function labels.

| **Example 2:** | |
|---|---|
| **Main Question** | How can you tell that this is a prokaryote or eukaryote cell? Choose one option from the following: A: It is a prokaryote because it doesn't have a nucleus B: It is a eukaryotic cell because it has a cell wall and a nucleus C: It is eukaryotic because it does not a nucleus nor a cell membrane D: It is prokaryote because it has a cell wall |
| **Answer** | B. |

| **Subquestion** | **Answer** | **Cognitive Label** |
|---|---|---|
| 1. What visible features can be observed in the cell image, such as the presence of boundaries or internal structures? | Rectangular cells with defined boundaries and distinct internal spots are visible. | Low-Level Vision Reception |
| 2. What do the observed features (rectangular shape, defined boundaries, distinct internal spots) correspond to in cellular biology? | Defined boundaries correspond to cell walls; internal spots correspond to nuclei. | Vision Knowledge Recall |
| 3. What cellular structures differentiate prokaryotic cells from eukaryotic cells? | Eukaryotic cells have nuclei and may have cell walls; prokaryotic cells lack nuclei. | Language Knowledge Recall |
| 4. Based on the cell's visible features and the definitions of prokaryotic and eukaryotic cells, which type of cell is shown in the image? | B: It is a eukaryotic cell. | Decision-Making |

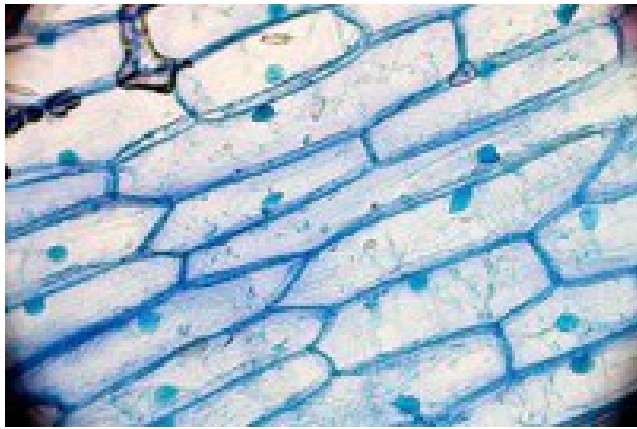

## A.4 COGVISION STATISTICS

The statistics for the CogVision dataset is shown in Table 8

Table 8: Dataset Statistics

| Metric | Training | Testing |
|---|---|---|
| Main Questions | 1,124 | 285 |
| Sub-questions | 4,603 | 1,141 |
| **Cognitive Skills Distribution** | | |
| High-Level Vision Reception | 1,262 (27.42%) | 336 (29.45%) |
| Math Reasoning | 570 (12.38%) | 125 (10.96%) |
| Semantic Understanding | 545 (11.84%) | 152 (13.32%) |
| Inference | 544 (11.82%) | 126 (11.04%) |
| Decision-Making | 454 (9.86%) | 111 (9.73%) |
| Language Knowledge Recall | 424 (9.21%) | 102 (8.94%) |
| Vision Knowledge Recall | 403 (8.76%) | 96 (8.41%) |
| Low-Level Vision Reception | 401 (8.71%) | 92 (8.06%) |

## A.5 THE COGNITIVE FUNCTION DISTRIBUTION OF OTHER MODELS

We present the heatmaps for the remaining five models in this subsection. The results reveal a notable universality in the sparsity patterns of attention heads across different architectures.

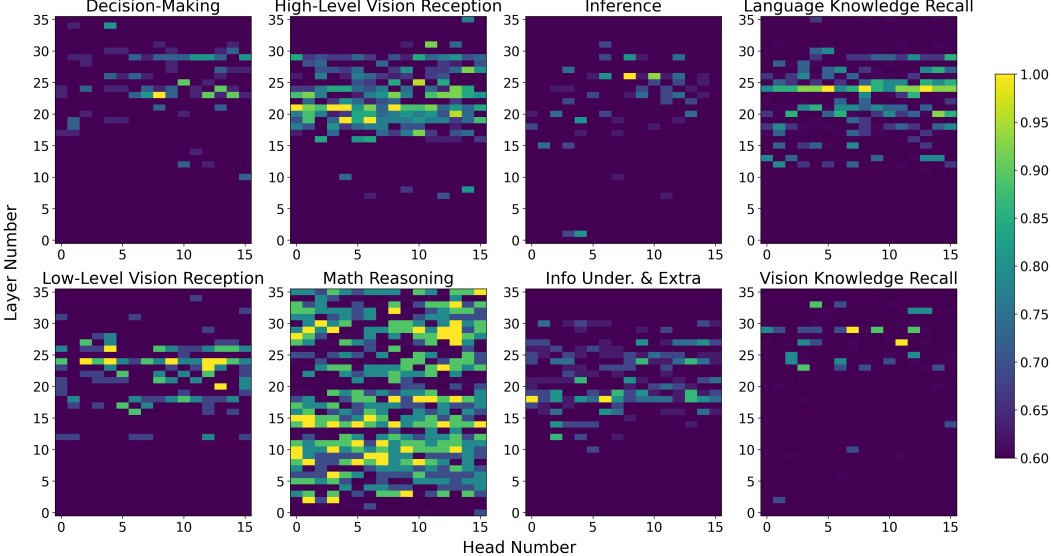

Figure 8: The existence of cognitive heads in Qwen2.5-VL-3B responsible for eight distinct functions in complex reasoning tasks. The x-axis represents the head index, while the y-axis indicates the layer index.

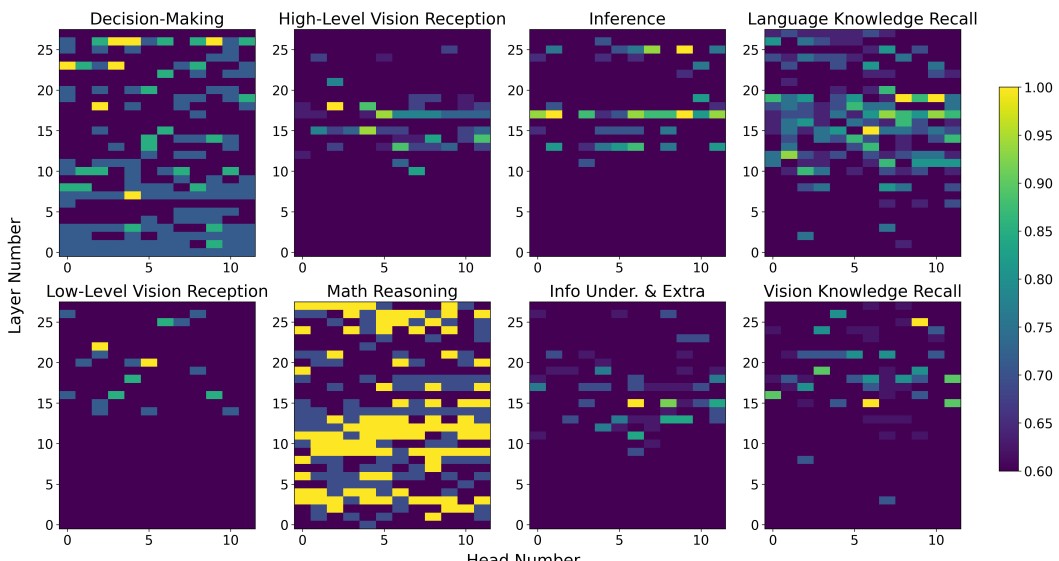

Figure 9: InternVL3-2B Heatmap

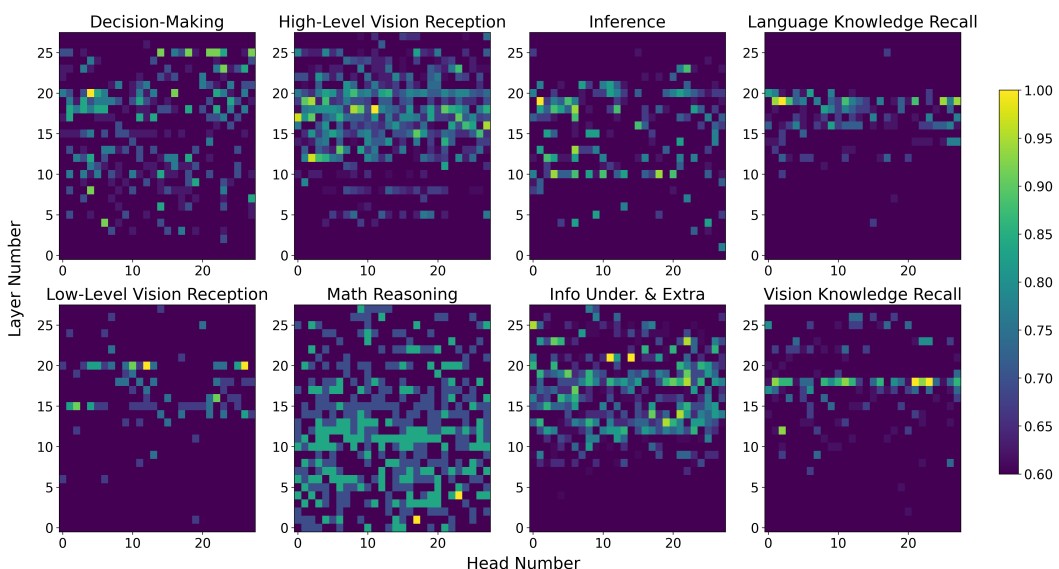

Figure 10: InternVL3-8B Heatmap

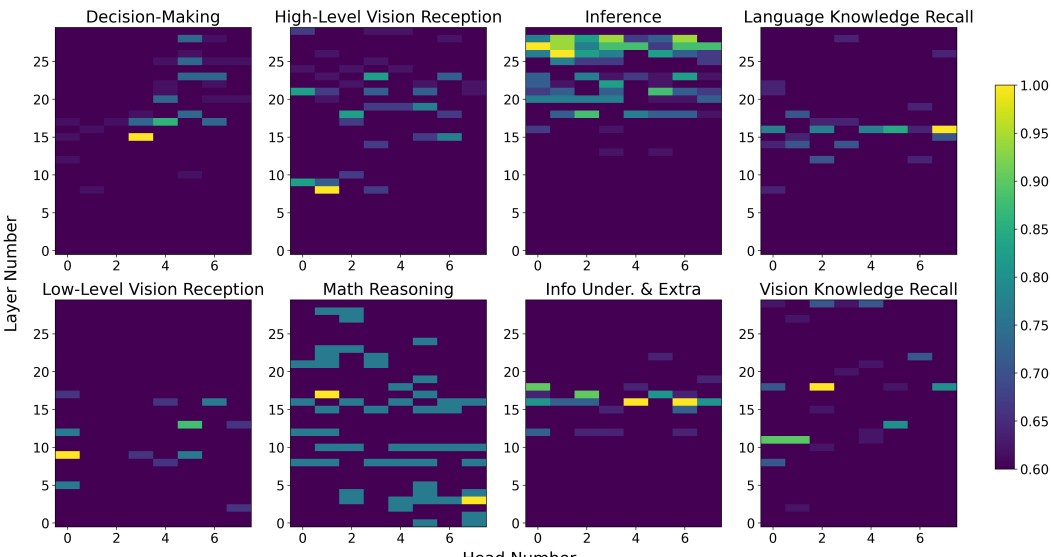

Figure 11: gemma-3n-e2b-it Heatmap

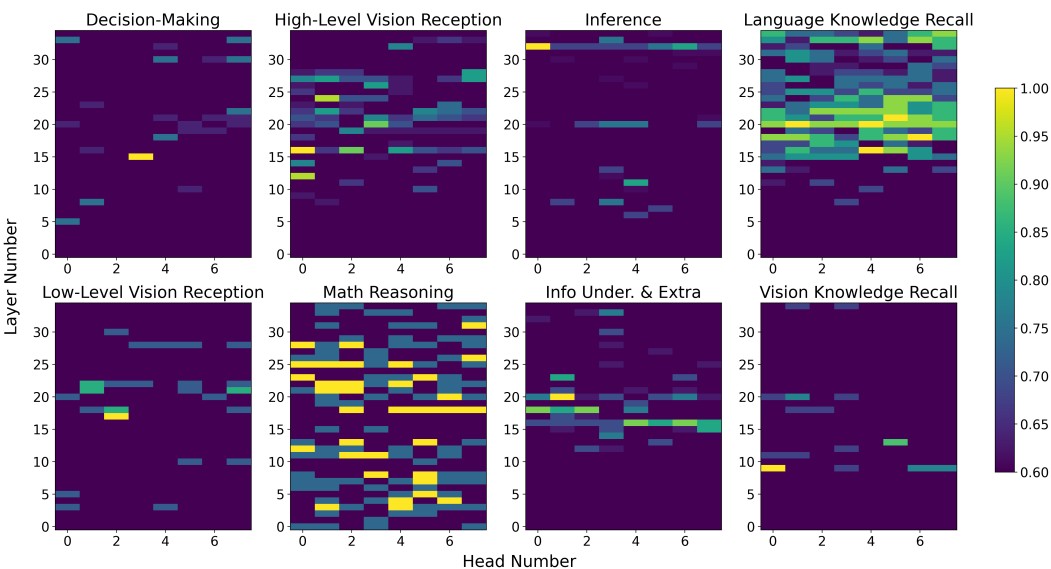

Figure 12: gemma-3n-e4b-it Heatmap

## A.6 ANNOTATIONS

To ensure the quality and reliability of the decomposed subQAF triplets in the CogVision dataset, we design a rigorous multi-stage annotation pipeline, combining expert review and model-based verification. The goal is to verify the logical validity of subquestions, the correctness of their associated cognitive function labels, and the accuracy of the answers. Notably, for each subquestion, we aim to align it with a single primary cognitive function. However, certain queries—such as "What is the solvent volume and how many particles in each solution?"—may involve multiple abilities (e.g., object recognition and counting). In such cases, we assign the subquestion to its dominant function in CogVision.

**Stage 1: Validating Subquestion Decomposition**  In the first stage, we evaluate whether the generated subquestions are logically sound and align with natural human reasoning. For each QA pair, three expert annotators (with backgrounds in linguistics or cognitive science) independently assess the validity of each subquestion. A subquestion is marked `true` if it meaningfully contributes to answering the main question and follows a logical reasoning trajectory. Otherwise, it is marked `false`.

We apply the following filtering criteria:

- **AI-Human Agreement**: If any annotator considers fewer than 60% of the subquestions valid, the entire QA decomposition is discarded.
- **Inter-Annotator Agreement**: A subquestion is deemed invalid if at least two annotators mark it as `false`. If over 40% of the subquestions in a QA pair are invalid under this rule, the whole QA pair is removed.

This filtering ensures that the retained QA decompositions follow coherent, cognitively plausible reasoning chains.

**Stage 2: Verifying Cognitive Function Labels**  In the second stage, annotators evaluate the correctness of the function label $f_i$ assigned to each subQAF triplet $(q_i, a_i, f_i)$. Three annotators independently mark each label as `true` or `false`. When discrepancies occur, annotators collaboratively reassign the correct cognitive label to ensure alignment with the underlying mental operation.

This step ensures that the categorization of subquestions accurately reflects established distinctions between information retrieval, semantic understanding, logical reasoning, and other cognitive processes.

**Stage 3: Answer Verification via Model and Human Review**  In the final stage, we verify the correctness of each answer $a_i$ using both automated and manual procedures. We employ the GPT-o3 model (OpenAI, 2024), known for its logical reasoning capabilities, to re-evaluate GPT-4.1-generated answers, and approximately 38.78% were found to be in disagreement. If GPT-o3 disagrees with GPT-4.1, it provides an alternative answer. A human annotator then compares both answers and resolves discrepancies by supplying the correct one when necessary. Given the generally objective nature of answers, only one annotator is required for this task.

**Annotation Outcome**  Following this multi-stage process, we retain 1,409 validated QA pairs, yielding a total of 5,744 high-quality subQAF triplets.

A.7  PROMPT FOR GENERATING COGVISION

We decompose the main question into subquestions through a two-step process: first, we prompt GPT-4.1 to generate a chain-of-thought (CoT) for the main question; second, we use the main question together with the CoT to guide the model in generating subquestions.

---

**Prompt**

**Generating CoT Prompt:**
You are an expert visual reasoning assistant. Given the following question and the correct answer, provide a detailed step-by-step chain of thought reasoning that leads to the correct answer. Here is the prompt for generating CoT:
Question: {question} Correct Answer: {correct_answer}
Please provide a detailed step-by-step analysis of how to solve this problem. Your response should: 1. Analyze what you see in the image 2. Break down the problem into logical steps 3. Conclude with the correct answer
Format your response as a clear paragraph, each step is one sentence.

---

Here is the prompt for generating subquestions:

**Prompt**

**Generating Subquestion Prompt:**
Here is the question:
<question>
{question}
<question>
Here is the chain-of-thought:
<chain-of-thought>
{cot}
<chain-of-thought>
Note
- Your task is to break the question down into detailed subquestions, ensuring each subquestion can be answered using only one specific cognitive skill.
- You need to create a structured and explicit reasoning process that simulates critical thinking while maintaining clarity and precision.
- The subquestion needs to be easy to answer and the answer needs to be concise
- The information of chain-of-thought cannot be used directly if it doesn't exist in main query.
- Each subquestion should be derived solely from the main query and the preceding subquestion.
- You CAN NOT retrieval information from chain-of-thought, but you can retrieval from question.
- Each subquestion should be designed to map to exactly one coginitive skill. - Your output should be formatted as a list of JSON objects, where each object represents a subquestion, its answer, and the required cognitive skill.
- You should use the most efficient logic to analyze the problem and minimize the number of subquestions.
Output format:

```
[
  {
    "subquestion": "<Subquestion text>",
    "answer": "<Concise answer>",
    "cognitive_skill": "<Assigned cognitive skill>"
  },
  {
    "subquestion": "<Subquestion text>",
    "answer": "<Concise answer>",
    "cognitive_skill": "<Assigned cognitive skill>"
  }
]
```

Your answer:

## A.8    PROMPT FOR QUESTION ASKING

---
**Prompt**

Here is the question:
<question>
{question}
<question>
Here is the prior knowledge in chain-of-thought (CoT) format:
<prior_knowledge>
{cot}
<prior_knowledge>
Instructions: - Answer the question carefully.
- You can use the information in prior_knowledge to help you answer the question.
- Your response should be clear and concise.
- Do not include any explanation, commentary, or code.
- Do not output anything after the closing square bracket.
Only output your final answer using this format:

```
[
    {{"answer": "<Your answer here>"}}
]
```

Your answer:

---

## A.9    ABLATION STUDY

In the main experiments, we use the top-k generated tokens and average their multi-head attention vectors. Here are some examples of selected top-k tokens by llm:

---
**Topk Tokens Selection Examples**

**Question:** What is the glaze on the chicken?
**Generated Full Answer:** The glaze on the chicken appears to be a light brown color with a glossy texture, suggesting it might be a sauce or marinade. The specific type of glaze cannot be determined from the image alone without more context.
**Ground Truth Answer:** The glaze is golden-brown, shiny, and smooth.
**Top 5 Tokens Selected:** glaze, light, brown, glossy, texture

---

---
**Topk Tokens Selection Examples**

**Question:** What is a reasonable estimate for the weight of this truck based on its classification?
**Generated Full Answer:** The weight of this truck is approximately 10,000 pounds (4,536 kilograms).
**Ground Truth Answer:** Around 2 tons.
**Top 5 Tokens Selected:** weight, approximately, 10,000, pounds, 4,536

---

We also explore alternative strategies for extracting representations, including using the first meaningful token, first token, last token and with or without layerbias. The corresponding results are shown in Table 9.

Sensitivity to the choice of k: We vary $k \in \{1, 3, 5\}$ and compute Pearson correlations of the resulting attention-head heatmaps. Figure 14 shows that the heatmaps remain highly correlated across choices of k, demonstrating that our method is robust to the exact number of selected tokens. This robustness arises because (1) the activation patterns associated with answering a subquestion are reflected across multiple output tokens, and (2) VLM outputs are short, reducing variance from token choice.

Table 9: Ablation experiment of topK tokens, first meaningful token, first token and last token, and with layerbias vs without layerbias on Qwen2.5-VL-3B-Instruct.

| Token | LayerBias | Inter_Head | Vision mainly Cognitive Functions | | | | | | Language mainly Cognitive Functions | | | | | | | | | |
|---|---|---|---|---|---|---|---|---|---|---|---|---|---|---|---|---|---|---|
| | | | Low-Level | | High-Level | | Recall | | Info | | Recall | | Math | | Inference | | Decision | |
| | | | llm | acc | llm | acc | llm | acc | llm | acc | llm | acc | llm | acc | llm | acc | llm | acc |
| First | without | random | 80.65 | 90.32 | 87.88 | 93.18 | 92.42 | 90.91 | 59.26 | 75.93 | 87.14 | 85.71 | 65.85 | 87.80 | 81.01 | 87.34 | 40.63 | 73.44 |
| | | cognitive | 48.39 | 54.84 | 45.24 | 48.23 | 80.30 | 83.33 | 50.00 | 72.22 | **0.00** | **0.01** | 82.93 | 95.12 | 55.70 | 63.29 | 64.06 | 70.31 |
| | with | random | 51.61 | 67.74 | 81.82 | 88.64 | 92.42 | 95.45 | 64.81 | 81.48 | 88.57 | 85.71 | 78.05 | 92.68 | 83.54 | 87.34 | 65.63 | 75.00 |
| | | cognitive | 45.16 | 48.39 | 56.06 | 66.67 | 86.36 | 86.36 | 48.15 | 62.96 | 74.29 | 81.43 | 78.05 | 90.24 | 51.90 | 56.96 | 68.75 | 75.00 |
| Last | without | random | 80.65 | 93.54 | 90.15 | 90.91 | 89.39 | 90.91 | 70.37 | 77.78 | 87.14 | 91.43 | 63.41 | 82.93 | 70.89 | 79.75 | 63.19 | 73.75 |
| | | cognitive | 12.90 | 12.90 | 66.94 | 67.12 | 86.36 | 86.36 | 38.89 | 64.81 | 15.71 | 18.57 | 82.93 | 95.12 | 6.33 | 7.59 | 48.44 | 50.00 |
| | with | random | 90.32 | 100.0 | 89.39 | 93.18 | 95.45 | 92.42 | 61.11 | 72.22 | 38.57 | 38.57 | 85.37 | 92.68 | 84.81 | 81.61 | 75.00 | 78.13 |
| | | cognitive | 67.74 | 64.52 | 89.39 | 93.18 | 67.74 | 64.52 | **0.06** | **0.02** | 68.57 | 64.29 | 82.93 | 95.12 | 43.04 | 46.84 | 65.63 | 70.31 |
| Meaning_first | without | random | 77.42 | 80.65 | 81.06 | 83.33 | 84.85 | 86.36 | 68.52 | 59.26 | 75.71 | 88.57 | 51.22 | 65.85 | 77.22 | 79.75 | 73.44 | 70.32 |
| | | cognitive | 77.42 | 87.10 | 50.76 | 54.55 | 84.85 | 75.76 | 50.00 | 53.70 | 2.85 | 4.23 | 85.37 | 95.68 | 72.15 | 79.75 | 57.81 | 75.00 |
| | with | random | 90.32 | 93.55 | 86.36 | 87.88 | 84.85 | 86.36 | 77.78 | 83.33 | 81.43 | 87.14 | 65.85 | 82.93 | 70.89 | 62.03 | 68.75 | 67.19 |
| | | cognitive | 67.74 | 64.52 | 68.18 | 72.73 | 84.85 | 75.76 | 59.26 | 62.96 | 2.85 | 4.23 | 7.31 | 9.76 | 72.15 | 79.75 | 48.44 | 57.82 |
| TopK | without | random | 83.87 | 90.32 | 82.58 | 83.33 | 84.85 | 86.36 | 64.81 | 70.37 | 87.14 | 94.29 | 78.05 | 92.68 | 58.23 | 91.14 | 78.13 | 90.63 |
| | | cognitive | 12.90 | 67.74 | 40.15 | 45.45 | **27.27** | **28.79** | 55.56 | 46.30 | 38.57 | 42.86 | 78.05 | 85.37 | 29.11 | 32.91 | 57.81 | 75.00 |
| | with | random | 87.10 | 96.77 | 82.58 | 83.33 | 86.36 | 84.85 | 59.26 | 55.56 | 85.71 | 85.71 | 82.93 | 87.80 | 91.14 | 86.08 | 81.25 | 82.81 |
| | | cognitive | **6.45** | **6.45** | **16.67** | **18.94** | 75.76 | 75.76 | 62.96 | 81.48 | 57.14 | 62.86 | **2.43** | **2.43** | **0.00** | **0.00** | 3.13 | 4.69 |

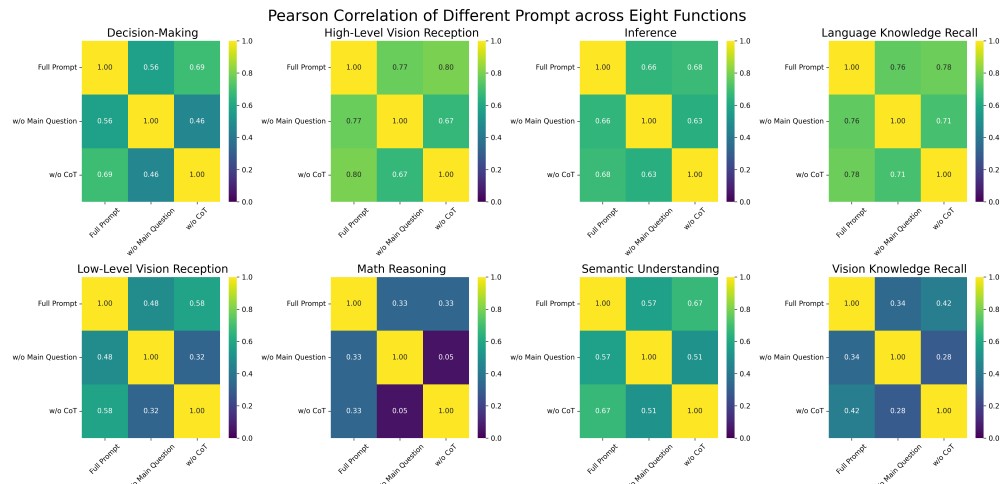

Figure 13: Pearson Correlation of different prompt types across eight functions for Qwen2.5-VL-3B-Instruct. Full Prompt: The prompt used in the main results, with given CoT, Main Question, Current Question. w/o Main Question: Prompt without main question involved.

Sensitivity to the choice of LLM: We further experimented with alternative LLMs for token selection. As shown in the examples in Table 10, different LLMs consistently select highly similar semantic tokens. The Pearson correlations of the resulting attention-head heatmaps (Figure 15) are likewise very high (almost 1), indicating that modern LLMs share a strong and consistent ability to identify key semantic units.

**Prompt format:** We also examined the influence of the main question and contextual input (preceding subquestions and their answers). Figure 13 shows that head importance maps vary noticeably across these changes, highlighting the importance of including both the main question and contextual information when identifying cognitive heads.

## A.10 MORE RESULTS

We conducted probing experiments using 3 random seeds. The results in Figure 16 demonstrate that our probing method is highly stable across seeds, with Pearson correlations of the heatmaps reaching 1 for all eight functions in Qwen2.5-VL-3B-Instruct.

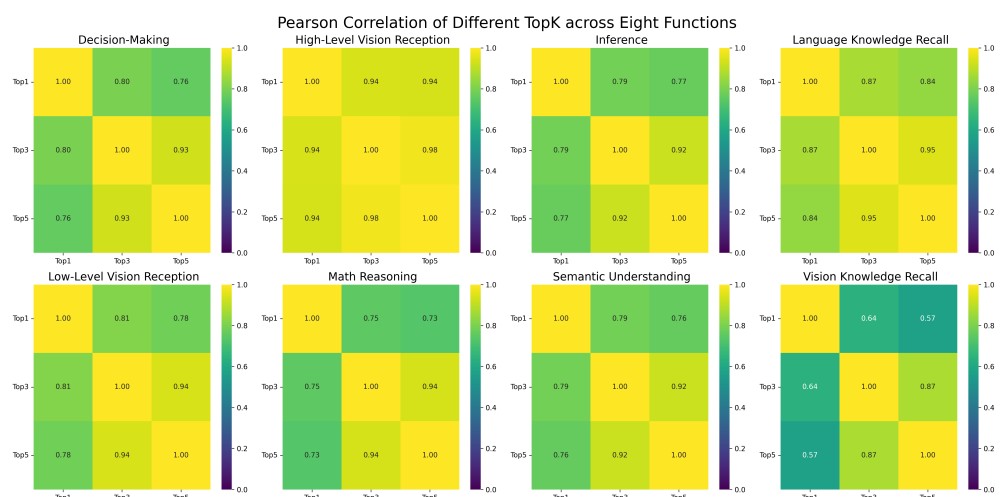

Figure 14: Pearson Correlation of different K of TopK token across eight functions for Qwen2.5-VL-3B-Instruct.

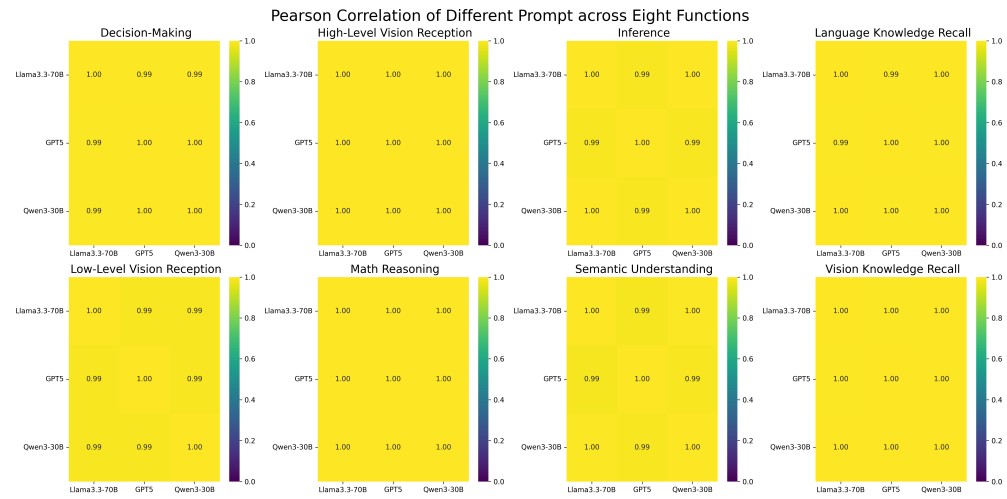

Figure 15: Pearson Correlation of different LLM selection across eight functions for Qwen2.5-VL-3B-Instruct.

Table 10: Examples of topk token selection using different LLMs.

| |
|---|
| Question: What are the notable visual features of the couch and love seat in the image, such as their shape, trim, and upholstery?
Answer: The couch and love seat in the image are patterned after a classic, elegant style with intrica te detailing and a neutral color palette. |
| LLM top5 token selection:
Qwen3-30B: classic, elegant, patterned, detailing, neutral
GPT5: classic, elegant, patterned, detailing, neutral
Llama3.3-70B: classic, elegant, patterned, detailing, color |
| Question: What familiar object matches the shape and features of the blue item shown in the image?
Answer: The blue item is a fire hydrant. |
| LLM top5 token selection:
Qwen3-30B: fire, hydrant
GPT5: fire, hydrant
Llama3.3-70B: fire, hydrant |

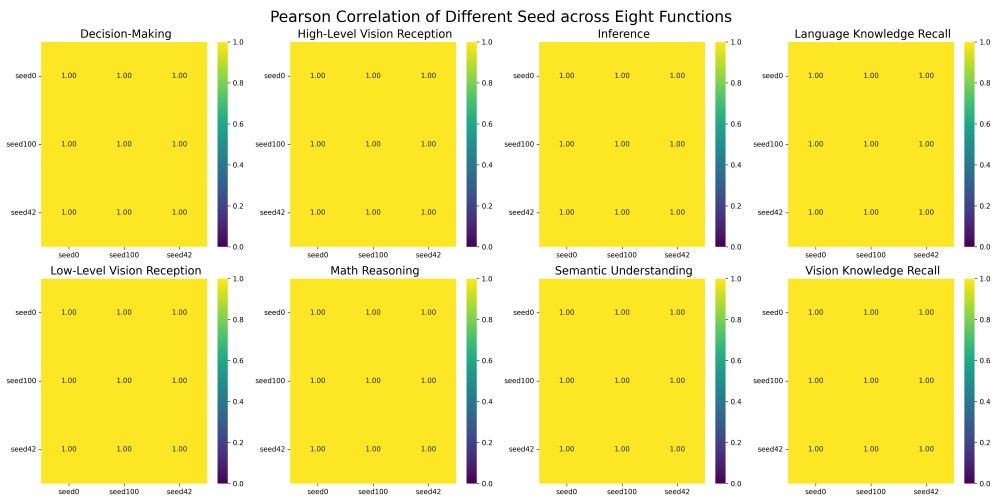

Figure 16: Pearson Correlation of different classification layer initialization seed across eight functions for Qwen2.5-VL-3B-Instruct.

**Sparse consistency across models:** The relatively high Pearson correlation coefficients between models in Figure 17 indicate that different models exhibit consistent sparsity patterns for different functions.

Table 11 reports the t-test results comparing random heads and cognitive heads, showing that the differences are statistically significant, (p-value≪0.05).

Table 11: Welch t-test between random heads and cognitive heads.

| Models | Vision mainly Cognitive Functions | | | Language mainly Cognitive Functions | | | | |
|---|---|---|---|---|---|---|---|---|
| | Low-Level | High-Level | Recall | Info | Recall | Math | Inference | Decision |
| | p-value (t-test) | p-value (t-test) | p-value (t-test) | p-value (t-test) | p-value (t-test) | p-value (t-test) | p-value (t-test) | p-value (t-test) |
| Qwen2.5-VL-3B-Instruct | 1.83e-03 | 5.71e-05 | 0.43 | 0.66 | 2.84e-03 | 4.11e-03 | 8.47e-06 | 1.39e-04 |
| Qwen2.5-VL-3B-Instruct | 8.27e-05 | 0.10 | 3.01e-03 | 2.91e-05 | 7.70e-03 | 0.76 | 0.46 | 7.71e-05 |
| InternVL3-2B | 0.02 | 0.05 | 8.58e-03 | 0.88 | 4.97e-03 | 1.49e-03 | 0.18 | 3.78e-03 |
| InternVL3-8B | 3.98e-07 | 4.86e-06 | 2.80e-04 | 3.15e-06 | 1.19e-03 | 1.86e-05 | 1.01e-03 | 0.19 |
| Gemma3-2B | 1.84e-06 | 0.19 | 2.32e-03 | 3.87e-03 | 1.98e-04 | 1.86e-04 | 9.43e-03 | 6.65e-05 |
| Gemma3-4B | 5.33e-05 | 0.04 | 0.01 | 0.07 | 2.28e-04 | 0.55 | 7.19e-03 | 4.15e-05 |

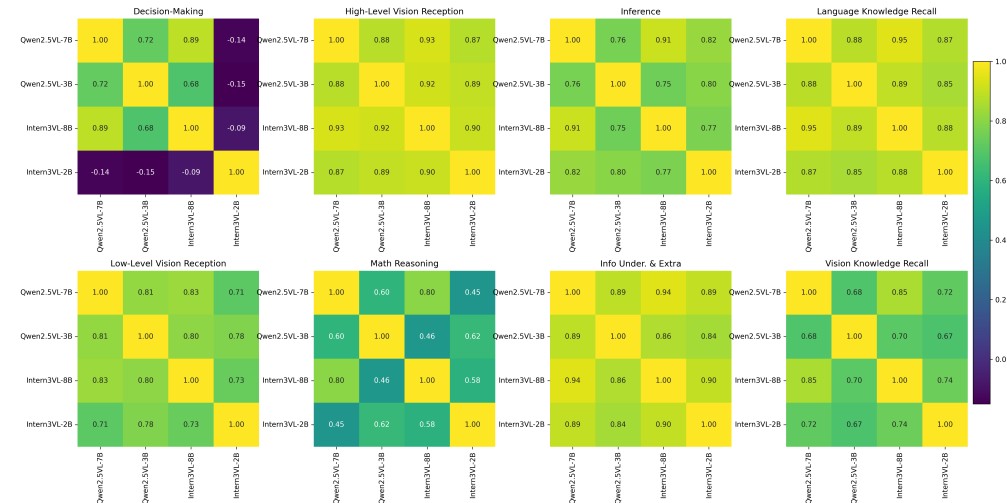

Figure 17: Pearson Correlation betweeen different models across eight functions.

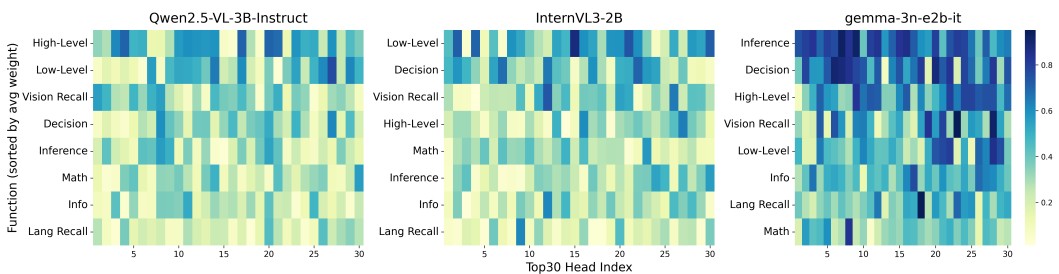

Figure 18: The importance (average attention weight) of visual modality for different functional heads (Top 30).

### A.11 DETAILS OF ACTIVATION PATCHING

In both the *random* and *mean* activation-patch methods, we replace the cognitive heads using patches from another function. To minimize correlation between functions, we use language-related functions when patching vision functions, and vice versa. Table 12 shows the corresponding function mappings.

Table 12: Function-to-Function Mapping Used for Activation Patching

| Original Function | Replacement Function |
|---|---|
| Vision Knowledge Recall | Decision-Making |
| Language Knowledge Recall | High-Level Vision Reception |
| Semantic Understanding | High-Level Vision Reception |
| Math Reasoning | Vision Knowledge Recall |
| Low-Level Vision Reception | Inference |
| Inference | Vision Knowledge Recall |
| High-Level Vision Reception | Semantic Understanding |
| Decision-Making | Vision Knowledge Recall |

A.12   NEGATIVE AND POSITIVE INTERVENTION CASES

The examples shown below illustrates a negative intervention case, where the model initially gives an correct answer but produces the incorrect one (The model fails to answer in this case) after intervention.

---

**Negative Intervention Case**

**Dataset:** CogVisioin (In-Domain)
**Query:** Drivers are instructed to stop their vehicles.
**Original Output:** Drivers are instructed to stop their vehicles at the 'STOP' sign.
**After negative Intervention (Semantic Understanding) Output:** Your answer here

---

The examples shown below illustrates the positive intervention success cases and failure case, where the model initially gives an incorrect answer (marked in red) but produces the correct one (marked in green) after intervention. This shows that adjusting the activation of cognitive heads along their corresponding functional directions enhances performance on visual reasoning tasks. However, in complex scenes, positive intervention may also have negative effects by assigning more weight to unimportant elements, for example, focusing on tools and materials while overlooking the toilet settings in the example below.

---

**Negative Intervention Case**

**Dataset:** Clevr-Math (Out-of-Domain) **Query:** You are a visual counting expert. Add 6 purple matte cylinders. How many purple matte cylinders are left?
**Ground Truth Answer:** 9
**Original Output:** Let's start by identifying the initial number of purple matte cylinders in the image. There are three purple matte cylinders visible: one small, one medium, and one large. Now, we need to add six more purple matte cylinders to the existing count. This means we will have: 1. The original small purple matte cylinder 2. The original medium purple matte cylinder 3. The original large purple matte cylinder 4. Six additional purple matte cylinders. Adding these up, we get: $1 + 1 + 1 + 6 = 9$. Therefore, the total number of purple matte cylinders after adding six more is 9.
**After negative Intervention (Math Reasoning) Output:** Let's start by identifying the initial number of purple matte cylinders in the scene: 1. There is one small purple matte cylinder on the left side of the image. 2. There is another small purple matte cylinder next to it. 3. There is another large purple matte cylinder. So adding them together, there are currently 2 purple matte cylinders. Now, let's add 6 more purple matte cylinders to the scene: 4. We will place these new matte cylinders around the existing ones. After adding the 6 new purple matte cylinders, we can count them: 5. The total number of purple matte cylinders now is 8. Therefore, the final answer is: 8

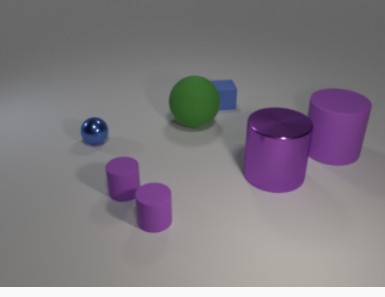

---

**Negative Intervention Case**

**Dataset:** okvqa (Out-of-Domain) **Query:** Which of the foods in the picture is best for you to eat when you have a cold?
**Ground Truth Answer:** soup
**Original Output:** In this image, there is a bowl of soup and a sandwich. Soup can be comforting and easy to digest, especially if it contains ingredients like vegetables or broth. However, the sandwich appears to be made with bread, which might be harder to digest for some people with a cold due to its high fiber content. Therefore, the food that is best for someone with a cold to eat would likely be the soup. Answer: The food in the picture that is best for you to eat when you have a cold is the soup.
**After negative Intervention (High-Level Visual Reception) Output:** The only food in the picture is a sandwich. Answer: Sandwich

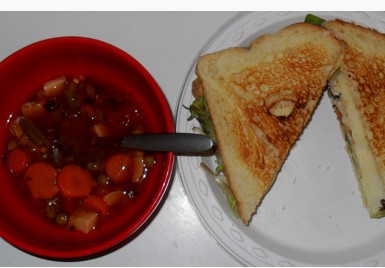

**Positive Intervention Case (Success Case)**

**Dataset:** CogVisioin (In-Domain)
**Query:** Based on their vibrant colors and textures, what are the designs in the grass most likely made out of?
**Ground Truth:** The designs are most likely made out of flowers.
**Original Output:** The designs in the grass are most likely made out of colored turf or artificial grass with embedded numbers and clock hands.
**Positive Intervention (High-Level Visual Reception) Output :** The designs in the grass are most likely made out of different types of grasses and flowers, possibly with some artificial elements like painted numbers for the clock face.

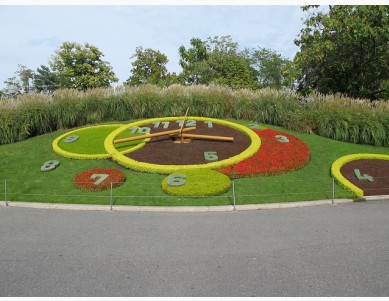

**Positive Intervention Case (Success Case)**

**Dataset:** MathVista (Out-of-Domain)
**Query:** You are a visual reasoning expert. What is the maximum value of this function?
**Ground Truth:** y = 1
**Original Output:** The function is represented by a parabola opening downwards. The vertex is at (2, 16). Therefore, the maximum value of the function is the y-coordinate of the vertex, which is 16.
**Positive Intervention (Math Reasoning) Output :** The image shows a graph of a function. The maximum value of the function is the highest y-value it reaches. Looking at the graph, the highest point is at y = 1.

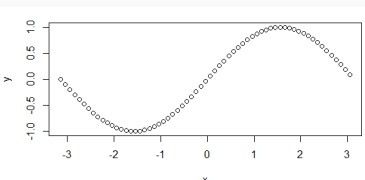

**Positive Intervention Case (Failure Case)**

**Dataset:** OK-VQA (Out-of-Domain)
**Query:** What type of person would you call if this happened in your house?
**Ground Truth:** Plumber
**Original Output:** The image shows a bathroom under construction or repair, with tools and a toilet missing its tank. A plumber is typically called for such tasks.
**Positive Intervention (High-Level Visual Reception) Output :** The image shows a bathroom under construction or renovation, with tools and materials scattered around. A handyman is a professional who performs repairs and maintenance tasks.

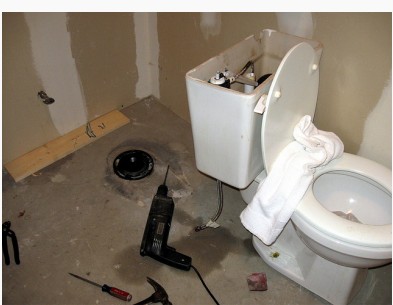

