# OpenReview forum: "Do Vision-Language Models Reason Like Humans? Exploring the Functional Roles of Attention Heads"
_ICLR.cc/2026/Conference — Submitted to ICLR 2026_

### Official Review · Reviewer_niyk · 2025-10-27

**Soundness:** 2
**Presentation:** 2
**Contribution:** 2
**Rating:** 2
**Confidence:** 3

**Summary:**

The paper investigates the role of attention heads for multimodal reasoning. Using a new dataset with multiple levels of questions, they try to map attention heads to human cognitive functions.

**Strengths:**

**Originality** The underlying question is interesting.

**Quality** The data set seems well motivated and the presentation is clear for the most part.

**Clarity** The goal of the paper is clear.

**Significance** The question of whether specific attention heads for subprocesses of multimodal reasoning exist in VLMs has not been explored as far as I know.

**Weaknesses:**

While the overall investigation seems well motivated, I think some of the take-aways are overstated. The writing is a bit sloppy at parts and should be improved. Some results are not explained very well. I appreciate taking inspiration from human cognitive processing, and I do think that there is more to be understood in regard to how well LLM and human cognitive processing relate to each other, but this paper does not present reliable insights in its current state. To summarize, I like the general idea but feel the paper is not in a state to be accepted.

**Questions:**

**Main questions:**
- While I think taking inspiration from human cognition is always nice, I do not think the abilities tested in the data subsets map as neatly onto brain areas as they are presented here. In the example in Figure 1 for example, the counting of particles in the respective solutions is surely something that other parts of the brain, such as the parietal lobe and the frontal lobe are involved with also. I see this whole approach as more of a guided questioning, where questions build on their predecessor.
- Line 153 for the data filtering and continued in the Appendix you write "A subquestion is deemed invalid if at least two annotators mark it as false. If over 40% of the subquestions in a QA pair are invalid under this rule, the whole QA pair is removed." What happens to subquestions that are deemed invalid? If two raters deem a single subquestion invalid, does it remain in the QA set anyways or is it removed while the other subquestions remain?
- Line 175 "To support coherent multi-step reasoning, we include preceding subquestions and their answers as contextual input" does this mean that even if a model does not correctly answer previous subquestions, it is queried on later subquestions given the correct answer as contextual input? I feel like this is kind of counterintuitive given your motivation of sequential processing. In this case, the model does not really perform sequential processing in a way that would be comparable to humans. Instead, it is given a very informative context with parts of the solution.
- Line 248 "These results demonstrate that VLMs rely on highly specialized, localized components for distinct cognitive abilities." I'm not sure this is a fair assessment, looking at Figure 2, there seems to be quite a lot of overlap in head activations between different abilities. In general, I think it'd be nice to compute a correlation between the activations compared to "accuracies over the eight functions". Also for this claim on the next line "Moreover, this sparse functional organization is consistent across architectures and scales: heatmaps for five additional model", I would want to see a correlation or consistency metric of sorts, rather than visual comparisons between the heatmaps. I think you go into this somewhat in line 369 with "8% of cognitive heads across eight functions participate in more than one function." but I don't know what exactly this means. I'd want you to clarify what participating in more than one function means here.
- For Figure 3 what is the difference between LLM and ACC lines? In line 307 it says "To quantify the impact, we employ both an LLM-based judge (Qwen3-30B LLM (Yang et al., 2025)) and an integrated accuracy metric". I understand the accuracy metric but what does the LLM based judge mean? Is the output of the masked model judged by this other LLM and then the fraction of answers judged as correct is shown here? I feel like this could use more detailed explanation. I'm similarly confused about Table 2, the numbers next to the arrows seem to indicate that performance before was at 100 and turning off specific heads reduced performance by this amount. Does this mean that for all analyses, you only analysed questions that the respective model got right before?
- Line 308, again explaining FIgure 3 "An output is considered unaffected if its BLEU score (Papineni et al., 2002) exceeds 0.8, or if either the ROUGE score (Chin-Yew, 2004) or the semantic similarity score surpasses 0.6." Which score then is reported in Figure 3? The y index only reads "score".
- Just so I understand correctly, in the masking analysis in section 4.2, you mask out the top % of heads that have the highest accuracies for a given subtask, not over all tasks, right? Am I right in thinking that Figure 4 shows the performance differences in relation to the unmasked model performance (basically, how much worse do the model outputs become for all of the different functions if you ablate the highest % for one specific function)?
- Line 367 "The neural system is inherently complex, with individual neurons often participating in multiple functions (Mante et al., 2013). We observe a similar phenomenon in VLMs" to me this feels like you are interpreting all results to mean the data fit humans. In line 248 you write "These results demonstrate that VLMs rely on highly specialized, localized components for distinct cognitive abilities" don't these two assessments clash?


**Minor comments**:
- Line 37 beginning should probably read "for a human" or "for humans"?
- LIne 51 you write "attention heads, an important component in VLMs". I think, since this paper is geared towards a machine learning crowd as well as the neuroscience crowd, it would be good to give some more explanation on what attention heads actually are.
- Line 53 "a dataset that bridges" bridges is missing the s, also Line 198 "triplets" are missing the s
- Line 106 you write "To systematically capture the cognitive processes involved in complex reasoning tasks, we consider eight functions related to complex multimodal reasoning, inspired by established frameworks in cognitive science (Anderson, 2014; Diamond, 2013)." Maybe you can give some more detail on how exactly the frameworks map to the eight functions you have chosen? This sentence reads a bit like these functions are generally taken to be fundamental in these cognitive frameworks, but as far as I remember ACT-R for example does not focus specifically on these.
- Line 200 "the attention-head output values that lead to true answers recognized as positive class while lead to other functions as negative class." is hard to understand and not grammatically correct
- Line 213 "(InternVL3-8B and InternVL3-8B)" should read 2B and 8B
- Line 284 "Intervention results (%) of cognitive heads" percent of correct answers?

---

> ### Author Response · Authors · 2025-11-21
>
> Dear Reviewer,
>
> Thank you for taking the time and effort to review our paper. We also appreciate your constructive comments on the writing, which have helped us improve the clarity and presentation of our work. Below, we address the questions you raised.
>
>
> **The writing should be improved. Some results need more explanation.**
>
> **A.** We appreciate your comments regarding writing clarity and explanation of results. In response, we have improved the manuscript by refining the writing, providing more detailed explanations, and conducting additional experiments to address your concerns. The details of these updates can be found in the following answers.
>
> **Q1. The abilities tested in the data subsets not map as neatly onto brain areas as they are presented here.**
>
> **A.** Drawing inspiration from the interaction between neural processes and human cognition, we propose a novel interpretability framework for systematically analyzing the roles and behaviors of attention heads in VLMs during reasoning. The example in Figure 1 illustrates that human brain engages multiple regions, each performing distinct cognitive functions. This motivates our investigation of functional roles of attention heads in VLMs.
>
> For each subquestion, we aim to align it with a single primary cognitive function. However, certain queries—such as “What is the solvent volume and how many particles in each solution?”—may involve multiple abilities (e.g., object recognition and counting). In such cases, we assign the subquestion to its dominant function in CogVision.
>
> Our work provides a first step toward exploring potential similarities between the cognitive processes of VLMs and those of the human brain, **without claiming a complete alignment**. As discussed in our paper, our approach has some limitations. We hope this opens a direction for future research. We will clarify this in the revised version. Furthermore, we revise our title to "Investigating The Functional Roles of Attention Heads in Vision Language Models: Evidence for Reasoning Modules" to prevent any implication of claiming full human-like reasoning.
>
> **Q2. What happens to subquestions that are deemed invalid? If two raters deem a single subquestion invalid, does it remain in the QA set anyways or is it removed while the other subquestions remain?**
>
> **A.**
> All subquestions removed. For example, if a main question has 10 subquestions and more than 4 are deemed invalid, all subquestions, whether valid or invalid, are removed, i.e., the whole QA pair are removed (as we illustrated in Appendix lines 924-926). This strategy is used to ensure that only QA pairs with coherent and reliable subquestion decompositions are retained.
>
> **Q3. Line 175 "To support coherent multi-step reasoning, we include preceding subquestions and their answers as contextual input" does this mean that even if a model does not correctly answer previous subquestions, it is queried on later subquestions given the correct answer as contextual input?**
>
> **A.** Thank you for pointing this out, we appreciate the opportunity to clarify.
>
> In our probing setup, the **correct subanswers** to preceding subquestions are indeed provided as contextual input. This design choice is essential for ensuring that the model can focus solely on the current subquestion, which is associated with a specific cognitive function, while having accurate prior information. This allows us to analyze the activation patterns of attention heads when the VLM processes the correct reasoning chain, enabling reliable probing results (see lines 199–201 in the paper).
>
> If we were to use the model’s own incorrect subanswers as context, two issues would arise:
> (1) the model would likely also produce an incorrect answer for the current subquestion, and then
> (2) the attention heads would fail to exhibit activation patterns aligned with the intended cognitive function, making the probing analysis unreliable.
>
> Furthermore, in the **Hierarchical Structure** (Subsection 4.3), we describe a complementary setting in which the model’s early-stage predictions are propagated to subsequent subquestions. This setting is used for structured behavioral analysis.

---

> ### Author Response · Authors · 2025-11-21
>
> Thanks for your patience.
>
> **Q4. Compute a correlation between the activations compared to "accuracies over the eight functions" and a correlation  between models. Clarify what participating in more than one function means here.**
>
> **A.** We clarify the claims and provide additional analyses as requested.
>
> **Correlation analyses across functions:** We computed Pearson correlations between head-activation heatmaps (derived from accuracy) across the eight functions. As shown in Figure 7 in [Rebuttal PDF](https://anonymous.4open.science/r/Anoymuous_ICLR-657E/VLM_function__ICLR_2026_Response_.pdf), the correlations are generally low, confirming that different functions tend to rely on partially separable head subsets.
>
> **Sparse consistency across models:** We have added Pearson correlation analysis to quantify the sparsity consistency across different models. As shown in Figure 8 in [Rebuttal PDF](https://anonymous.4open.science/r/Anoymuous_ICLR-657E/VLM_function__ICLR_2026_Response_.pdf), the relatively high Pearson correlation coefficients between models indicate that different models exhibit consistent sparsity patterns for different functions. This aligns with the heatmap visualizations, where a small set of prominently activated heads appears across architectures and scales.
>
> Finally, regarding the statement “18\% of cognitive heads participate in more than one function”: Here, participate refers to heads that exceed the threshold (top 10\%) for multiple cognitive functions in our probing analysis. In other words, these are heads that appear in the top-activated set for more than one function.
>
> **Q5. For Figure 3 what is the difference between LLM and ACC lines? Is the output of the masked model judged by this other LLM and then the fraction of answers judged as correct is shown here? Does this mean that for all analyses, you only analysed questions that the respective model got right before?**
>
> **A.** 1.In Figure 3, “TopK LLM” and “RandomK LLM” correspond to the LLM-judge metric, while “TopK ACC” and “RandomK ACC” correspond to the integrated accuracy metric.
>
> 2. That is correct: we use a separate LLM (Qwen3-30B) to judge the correctness of answers. LLM-based judging has been shown to be effective and is commonly used in recent studies [1–2]. Its effectiveness is also reflected in our results, as the LLM lines closely overlap with the ACC lines.
>
> 3. It is right, i.e., we only evaluate subquestions that the model originally answered correctly. This filtering ensures that any observed drop in performance is caused solely by the intervention.
>
> [1] Wang, Xinran, et al. "Map: Multi-human-value alignment palette." ICLR 2025.
>
> [2] Liu, Shuliang, et al. "Judge as a judge: Improving the evaluation of retrieval-augmented generation through the judge-consistency of large language models." Findings of the Association for Computational Linguistics: ACL 2025.
>
> **Q6. Which score then is reported in Figure 3?**
>
> **A.** In Figure 3, the blue line represents the LLM-based judge, while the red line shows the integrated accuracy metric (as indicated in the legend). The integrated accuracy combines the BLEU score, ROUGE score, and semantic similarity score to provide a comprehensive evaluation of the output.
>
> **Q7. In the masking analysis in section 4.2, you mask out the top \% of heads that have the highest accuracies for a given subtask, not over all tasks, right? Am I right in thinking that Figure 4 shows the performance differences in relation to the unmasked model performance (basically, how much worse do the model outputs become for all of the different functions if you ablate the highest \% for one specific function)?**
>
> **A.** Your understanding is correct. In subsection 4.2, we mask out the top \% of heads that have the highest accuracies for a given subtask, not over all tasks. In Figure 4, we mask heads associated with one function (e.g., language knowledge recall) while evaluating performance on all subtasks.

---

> > ### Author Response · Authors · 2025-11-21
> >
> > Thanks for you patience.
> >
> > **Q8. Don't these two assessments (highly specialized heads and heads across functions)  clash?**
> >
> > **A.** We would like to clarify that these two observations are not contradictory.
> >
> > 1. Highly specialized heads for each cognitive function are verified by the sparse cognitive heads in Figure 2 (with additional heatmaps in the Appendix), and their functional contributions are shown in Table 1 and Figure 4.
> >
> > 2. In our approach, we quantify and rank the accuracy of attention heads for each cognitive function. Heads with high accuracy scores are identified as cognitive heads for that function. **Thus, with our probing-based method, a head that ranks highly for one function may also exhibit non-negligible importance for others, leading to the phenomenon of "Heads Across Functions."**
> > Notably, even if a head ranks in the top 10\% for multiple cognitive functions, our ranking still reveals a primary function for which it is most diagnostic.
> >
> > 3. Large models like VLMs are built on neural networks inspired by biological neural systems, so their inherent complexity, which resembles that of the human brain, is reasonable, but these observations are not intended to claim a full alignment with human neurobiology, we will highlight this.
> >
> > **For Other Minor comments**
> >
> > **A.** Thanks for your insightful comments. We have revised the typos and clarified the text accordingly. Specifically:
> >
> > Line 37: updated to “for humans.”
> >
> > Line 51: added a brief explanation of attention heads as key transformer units computing token relationships.
> >
> > Line 53 / 198: corrected “bridges” and “triplets.”
> >
> > Line 106: clarified how the eight cognitive functions are derived from cognitive frameworks and adapted for multimodal reasoning.
> >
> > Line 200: rephrased for clarity.
> >
> > Line 213: corrected model names to InternVL3-2B and InternVL3-8B.
> >
> > Line 284: clarified that intervention results (\%) refer to the fraction of correct answers.

---

> > > ### Comment · Reviewer_niyk · 2025-11-25
> > > **Reminder to update the manuscript on OpenReview**
> > >
> > > Dear authors,
> > >
> > > Thank you for your extensive rebuttal response. Could you also upload the updated manuscript to openreview so we can check the changes made to the manuscript? I will then respond to the individual points.

---

> ### Author Response · Authors · 2025-11-26
>
> Dear Reviewer,
>
> We are grateful for your response. In the updated manuscript (with changes highlighted in blue), we have incorporated your suggestions as follows:
>
> - We revised the title to “Investigating the Functional Roles of Attention Heads in Vision-Language Models: Evidence for Reasoning Modules”. We also thoroughly reviewed the manuscript and clarified in the Conclusion (Lines 531–534) that our work does not claim that VLMs perform full human-like reasoning.
>
> - We added correlation analyses across cognitive functions (Lines 255–257) and across models (Lines 259–260), with detailed results provided in Figure 17 of Appendix A.10.
>
> - More explanations about LLM-based judge (lines 308-309), "Heads Across Functions" (lines 418-422), Table 2 (lines 395-396), and clarification of limitations on CogVision (lines 1022-1025).
>
> We carefully reviewed the entire manuscript to correct typographical and grammatical errors.
> We sincerely appreciate your helpful comments, and we hope that our revisions address your concerns. We would be grateful for any further suggestions or discussion.
>
> Best Regards,
>
> The Authors

---

> > ### Comment · Reviewer_niyk · 2025-11-27
> > **Reviewer reply to rebuttal response**
> >
> > Dear authors,
> >
> > first off, I really appreciate the extensive rebuttal and the changes to the writing and the title. I will list my remaining concerns below:
> >
> > For Q2, I'm not sure this. completely answers my question. You say "For example, if a main question has 10 subquestions and more than 4 are deemed invalid, all subquestions, whether valid or invalid, are removed", so the threshold is 4 or is a whole QA pair already removed if human reviewers deem a single subquestion invalid?
> >
> > For Q3, you write "Furthermore, in the Hierarchical Structure (Subsection 4.3), we describe a complementary setting in which the model’s early-stage predictions are propagated to subsequent subquestions. This setting is used for structured behavioral analysis.", just to make sure, this means that the results in Table 3 are obtained using the models self-generated output as input for later subquestions, whereas otherwise in the paper you use the ground truth answers for previous subquestions as input? I feel this could still be made more clear.
> >
> > For Q4 and the sparsity Figure 8 in the Appendix. I appreciate you having run these analyses. If you add these to the Appendix of the updated manuscript I suggest you fix the cmap limits of all subplots so they are easily comparable.
> >
> > For Q8, I understand that with your methodology you can have sparse heads for each function but that these heads may also be active for multiple different functions. However, the my point was more about the framing and how I felt you were at parts taking contrasting results to both mean a good fit to human data. In line 83 you write "correlations among these functional heads reveals cross-function interaction, where a single head may support multiple functions or modalities", I feel this is at odds with the functional specialization you suggest that cognitive abilities have in humans (math reasoning in the parietal lobe).
> >
> > I do like the genereal idea of testing specific cognitive functions and comparing human and model behavior and I appreciate the work put into the rebuttal and am therefore raising my score. However, in general I retain some concerns with the framing of the work. I feel it presupposes a too locally specialized nature for human cognitive functions, it selects a not entirely obvious subset of cognitive functions and seemingly contrasting results (at least to me) are taken to generally support alignment with human cognition.

---

> > > ### Author Response · Authors · 2025-12-02
> > >
> > > Dear Reviewer,
> > >
> > > Thank you very much for your constructive feedback and for raising your score. We sincerely appreciate the time and care you have put into reading our rebuttal and manuscript revisions. Below we respond to each of your remaining concerns.
> > >
> > > **1. For Q2: Clarification of the Filtering Procedure**
> > >
> > > In CogVision annotation, each subquestion is labeled as True or False by three independent annotators.
> > >
> > > Our filtering procedure applies two criteria:
> > >
> > > **AI-Human Agreement**: If **any annotator** considers fewer than **60\% of the subquestions with True**, the entire QA decomposition is discarded.
> > >
> > > **Inter-Annotator Agreement**: A subquestion is deemed invalid if at least two annotators mark it as **False**. If **over 40\% of the subquestions in a QA pair are invalid** under this rule, the whole QA pair is removed. In other words, we first determine the validity of each subquestion based on annotator agreement and then apply a **60\% validity threshold** at the QA-pair level. The given example "For example, if a main question has 10 subquestions and more than 4 are deemed invalid, all subquestions, whether valid or invalid, are removed" satisfies this rule.
> > >
> > > The details are provided in Appendix A.6 (lines 1032–1040).
> > >
> > > **2. For Q3: Clarification for Table 3**
> > >
> > > Your are right that in Table 3, we use the model’s own generated output as input for subsequent subquestions, since our goal is to observe how early-stage cognitive functions influence later reasoning steps. In all other CogVision evaluations, we use the ground-truth answers. Moreover, for out-of-domain downstream tasks, no subquestions are involved, and the model directly produces the final output.
> > >
> > > **3. For Q4: Making subplots easily comparable**
> > >
> > > Thank you for this helpful suggestion.
> > > We have already fixed all colormap limits across subplots to ensure direct comparability (Figure 17 in Appendix A.10).
> > >
> > > **4. For Q8: Functional Specialization and Cross-Function Interaction**
> > >
> > > We have revised the framing throughout the manuscript to avoid potential confusion.
> > >
> > > To clarify, our notion of functional specialization refers to the following:
> > >
> > > - **Sparsity:** only a small subset of heads is strongly activated for each function (i.e., achieving high accuracy or high rank);
> > >
> > > - **Functional contribution:** demonstrated by Table 1 and Figure 4;
> > >
> > > - **Low cross-function correlations:** as shown in Figure 17 in Appendix A.10.
> > >
> > > We highlight that we do not claim that these heads exhibit full human-level anatomical specialization (e.g., “math reasoning in the parietal lobe”) in Conclusion.
> > >
> > > We sincerely thank you for acknowledging the contributions. Your comments significantly improved the clarity and framing of our work.
> > >
> > > Sincerely,
> > >
> > > The Authors

---

### Official Review · Reviewer_idDf · 2025-10-29

**Soundness:** 3
**Presentation:** 2
**Contribution:** 2
**Rating:** 4
**Confidence:** 4

**Summary:**

The authors built a dataset and a probing+intervention framework to show that specific attention heads in VLMs act like “functional heads” (e.g., math, visual reception, decision), are sparse and organized, and causally matter for multimodal reasoning. The authors probe VLM attention heads by extracting per-head activations from token-selected answer traces and training simple probes to score each head’s association with a function. The authors then run interventions: (i) negative—masking/attenuating selected heads—and (ii) positive—nudging heads along learned “functional directions.” Experiments span three VLM families. The work suggests VLMs contain function-specialized, causally relevant mechanisms that can be studied and steered, opening doors to interpretability and targeted test-time control.

**Strengths:**

1) Head-level masking and directional “positive interventions” move beyond correlational probing and show functional necessity/sufficiency signals.
2) Results replicated on InternVL, Qwen-VL, and Gemma at multiple sizes which mean claims aren’t model-specific.
3) Shows where (layers) and on what (image vs text tokens) different functions concentrate, plus cross-modal bridge heads.
4) Interventions on function-specific heads affect OK-VQA/Clevr-Math, suggesting external validity beyond the in-domain dataset.

One thing I love the most is its robust use of control experiments to demonstrate that the identified cognitive heads are function-specific and not random artifacts. The authors repeatedly compare the impact of masking their identified functional heads versus masking an equal number of randomly selected heads. In Figure 3, for instance, they show that masking up to 10% of random heads in Qwen2.5-VL-3B has negligible effect on performance, whereas masking the same fraction of cognitive heads leads to a steep drop in accuracy—underscoring their causal importance. Table 1 reinforces this across six models and eight functions, where performance on the corresponding cognitive skill sometimes drops to near-zero after masking, while random masking leads to minor degradation. They further strengthen this claim via cross-function controls in Figure 4, where masking heads from unrelated functions causes far less harm than masking those directly associated with the function under evaluation. They need to do statistical tests like human psychology control experiments however.

**Weaknesses:**

1. The paper does not appear to report standard statistical significance testing (e.g., t-tests, ANOVA, or confidence intervals) for its main results. The paper relies primarily on descriptive accuracy comparisons—such as consistent drops/improvements in Figure 3, Table 1, and Table 4—rather than formal statistical analysis. It supports claims through visual inspection (e.g., attention heatmaps in Figures 2 and 5–9) and contrasts between cognitive vs. random head masking.
2. The authors claim to investigate "cognitive" side of MLLM but have insufficient engagement of cognitive vision in MLLM literature such as,

Schulze Buschoff, L. M., Akata, E., Bethge, M., & Schulz, E. (2025). Visual cognition in multimodal large language models. Nature Machine Intelligence, 7(1), 96-106

Li, Y., Gao, Q., Zhao, T., Wang, B., Sun, H., Lyu, H., ... & Deng, H. Core Knowledge Deficits in Multi-Modal Language Models. In Forty-second International Conference on Machine Learning.

minor
1. Uniform scaling (ε) can cause off-target side effects; no comparison to alternative causal methods (path patching, causal tracing, activation patching).
2. Linear probes on selected tokens may conflate feature presence with linear separability; top-k token selection itself is LLM-driven.
3. Focus is only on attention heads; MLPs, attention patterns (Q/K), and routing components aren’t analyzed; limited stats on variance/seed stability.

**Questions:**

1) How consistent are subquestion function labels across annotators, and how often did GPT-o3 disagree?
2) How sensitive are head importance maps to (a) top-k token choice, (b) prompt format/order, and (c) random seeds across probes/interventions?
3) Do you think analogous “functional directions” exist in MLP blocks or cross-attention vs self-attention splits, and does combining head+MLP interventions yield larger, more stable gains?

---

> ### Author Response · Authors · 2025-11-21
>
> Dear Reviewer,
>
> We deeply appreciate the time and effort you have dedicated to reviewing our work. Below, we address the questions you raised.
>
> **Q1. The paper does not appear to report standard statistical significance testing (e.g., t-tests, ANOVA, or confidence intervals) for its main results.**
>
> **A.** The results in Table 1 show that the performance differences between random masking and cognitive masking are substantial, with gaps of tens of percentage points. These differences clearly demonstrate the functional contribution of the identified cognitive heads.
>
> To further strengthen our analysis, we additionally run masking experiments with 5 random seeds and include standard statistical significance tests in the revised version. The results in the following table demonstrate that all findings are statistically significant (p-value$<$0.05).
>
> Table. Multiple Random-Head Masking: Welch t-test
>
> | Models                     | Low-Level (Vision) | High-Level (Vision) | Recall (Vision) | Info (Language) | Recall (Language) | Math | Inference | Decision |
> |----------------------------|-----------------|------------------|----------------|----------------|-----------------|------|-----------|---------|
> | Qwen2.5-VL-3B-Instruct     | 1.83e-03        | 5.71e-05         | 0.43           | 0.66           | 2.84e-03        | 4.11e-03 | 8.47e-06  | 1.39e-04 |
> | Qwen2.5-VL-3B-Instruct     | 8.27e-05        | 0.10             | 3.01e-03       | 2.91e-05       | 7.70e-03        | 0.76     | 0.46      | 7.71e-05 |
> | InternVL3-2B               | 0.02            | 0.05             | 8.58e-03       | 0.88           | 4.97e-03        | 1.49e-03 | 0.18      | 3.78e-03 |
> | InternVL3-8B               | 3.98e-07        | 4.86e-06         | 2.80e-04       | 3.15e-06       | 1.19e-03        | 1.86e-05 | 1.01e-03  | 0.19     |
> | Gemma3-2B                  | 1.84e-06        | 0.19             | 2.32e-03       | 3.87e-03       | 1.98e-04        | 1.86e-04 | 9.43e-03  | 6.65e-05 |
> | Gemma3-4B                  | 5.33e-05        | 0.04             | 0.01           | 0.07           | 2.28e-04        | 0.55     | 7.19e-03  | 4.15e-05 |
>
>
> **Q2. More related works about cognitive vision in MLLM.**
>
> We thank the reviewer for introducing these important related works [1-2]. We have carefully reviewed them and incorporated their discussions into the revised manuscript.
>
> [1] Schulze Buschoff, L. M., Akata, E., Bethge, M., and Schulz, E. (2025). Visual cognition in multimodal large language models. Nature Machine Intelligence, 7(1), 96-106
>
> [2] Li, Y., Gao, Q., Zhao, T., Wang, B., Sun, H., Lyu, H., ... and Deng, H. Core Knowledge Deficits in Multi-Modal Language Models. In Forty-second International Conference on Machine Learning.
>
> **Q3. Uniform scaling ($\epsilon$) can cause off-target side effects; no comparison to alternative causal methods (path patching, causal tracing, activation patching).**
>
> **A.** Masking related important heads or random heads (e.g., scaling their activations with a small value) is a standard method to validate the functional role of identified heads [3–4]. Applying a similar suppression to the cognitive heads, we observe substantial performance drops when masking the identified cognitive heads, whereas masking random heads produces only minor effects, strongly suggesting that these effects are not accidental or side artifacts.
>
> We additionally conduct activation patching experiments, in which the activations of cognitive heads associated with one function are replaced by those from another function using two strategies: random and mean. In the random setting, activations are substituted with those from a randomly selected question belonging to a different function. In the mean setting, activations are replaced with the average activation computed over all questions associated with that function. As shown in following Table, **both types of activation patching result in substantial performance degradation, consistent with the effects observed under scaling interventions**.
>
> Table. Ablation Study of Different Activation-Masking Methods (LLM judge).
>
> | Token   | Inter_Head | Vision Low-Level | Vision High-Level | Vision Recall | Language Info | Language Recall | Math | Inference | Decision |
> |-|-|-|-|-|-|-|-|-|-|
> |  Base  | random| 87.10 | 82.58  | 86.36 | 59.26  | 85.71  | 82.93  | 91.14  | 81.25  |
> | Random  | cognitive  | **0.00**  | 20.45  | **62.12** | 35.79 | 12.86 | 17.07         | 14.32           | 6.25            |
> | Mean    | cognitive  | 3.80   | 17.39    | 66.67         | **35.19**     | **10.00**      | 37.17         | 3.80            | 6.25            |
> | Scalar  | cognitive  | 6.45  | **16.67**       | 75.76         | 62.96         | 57.14          | **2.43**      | **0.00**        | **3.13**        |
>
> [3] Wu, Wenhao, et al. "Retrieval head mechanistically explains long-context factuality." ICLR 2025.
>
> [4]Zhou, Zhenhong, et al. "On the role of attention heads in large language model safety." ICLR 2025.

---

> ### Author Response · Authors · 2025-11-21
>
> Thanks for you patience.
>
> **Q4. Linear probes on selected tokens may conflate feature presence with linear separability; top-k token selection itself is LLM-driven.**
>
> **A.**
> 1. **The effectiveness of top-k token selection:**
> We prompt Qwen3-30B to select the five most semantically important tokens in the predicted answer.
> The **examples in Appendix A.8** show that the selected tokens can semantically represent the full output.
>
> Table 8 in Appendix A.8 shows the effectiveness of top-k token compared to first meaning token. In the following table, we present complement results of masking cognitive heads based on different token selection strategies (the first token is used in previous work [1]): first is the first token, last is the last token, meaning_first is the first meaning token (excluding formatting), top-k is the top-k most semantically important tokens. We observe that top-k token masking leads to the most significant performance drop when masking the cognitive heads, indicating higher precision in identifying different cognitive heads.
>
>
> Table. Ablation experiment of topK tokens, first meaningful token, first token and last token on Qwen2.5-VL-3B-Instruct, using LLM-judge.
> | Token  | Inter_Head | Vision Low-Level | Vision High-Level | Vision Recall | Language Info | Language Recall | Math | Inference | Decision |
> |-|-|-|-|-|-|-|-|-|-|
> | First | random | 51.61 | 81.82 | 92.42  | 64.81  | 88.57  | 78.05 | 83.54 | 65.63 |
> | First | cognitive  | 45.16  | 56.06  | 86.36  | 48.15  | 74.29  | 78.05 | 51.90 | 68.75  |
> | Last  | random | 90.32 | 89.39 | 95.45  | 61.11 | 38.57 | 85.37  | 84.81  | 75.00  |
> | Last | cognitive  | 67.74  | 89.39 | 67.74  | 0.06  | 68.57  | 82.93 | 43.04 | 65.63 |
> | Meaning_first | random | 90.32   | 86.36   | 84.85 | 77.78 | 81.43  | 65.85 | 70.89 | 68.75 |
> | Meaning_first | cognitive  | 67.74 | 68.18  | 84.85  | 59.26 | 2.85 | 7.31 | 72.15 | 48.44 |
> | TopK  | random | 87.10  | 82.58  | 86.36 | 59.26 | 85.71 | 82.93 | 91.14  | 81.25  |
> | TopK  |cognitive  | **6.45**  | **16.67**| 75.76| 62.96 | 57.14 | **2.43** | **0.00**  | **3.13**  |
>
> We further experimented with alternative LLMs for token selection. As shown in the examples (added to the Appendix in the revised version), different LLMs consistently select highly similar semantic tokens, reflecting that modern LLMs share strong capability in identifying key semantic units.
>
> Question: What are the notable visual features of the couch and love seat in the image, such as their shape, trim, and upholstery?
>
> Answer: The couch and love seat in the image are patterned after a classic, elegant style with intrica te detailing and a neutral color palette.
>
> LLM top5 token selection:
>
> **Qwen3-30B: classic, elegant, patterned, detailing, neutral**
>
> **GPT5: classic, elegant, patterned, detailing, neutral**
>
> **Llama3.3-70B: classic, elegant, patterned, detailing, color**
>
> 2. **Causal evidence from different aspects:** (1). **Subsection "Functional Contribution of Cognitive Heads"** shows that masking cognitive heads lead to a
> substantial decline in performance, whereas masking an equal number of random heads results in only minor degradation across all VLMs. Our newly added activation patching experiments further corroborate these findings: replacing the activations of cognitive heads with those from another function—whether using random patching or mean patching—produces clear and consistent performance drops, reinforcing that these heads play causal, function-specific roles.
> (2). **Figure 4 in the paper further validate their functional roles**: masking the relevant functional heads (e.g., language knowledge recall heads for language knowledge recall task) yields a significantly larger performance drop than masking unrelated heads (e.g., vision knowledge recall), confirming their functional specialization.
> (3). **Downstream task interventions provide additional evidence.**
> Negative and positive intervention experiments, as illustrated in Appendix A.10,
> show that manipulating cognitive heads selectively affects related functions, further supporting their causal and functional roles.
>
> [1] Zhou, Zhenhong, et al. "On the role of attention heads in large language model safety." ICLR 2025.

---

> > ### Author Response · Authors · 2025-11-21
> >
> > Thanks for your patience.
> >
> > **Q5. Focus is only on attention heads; MLPs, attention patterns (Q/K), and routing components aren’t analyzed; limited stats on variance/seed stability.**
> >
> > **A.** 1. Our work specifically targets the functional roles of attention heads because they provide a clear, interpretable unit of analysis for understanding cognitive functions in VLMs. While other components like MLPs are also important, we leave their detailed analysis to future work (as we illustrated in **Limitations** in the paper).
> >
> > 2. We conducted experiments using multiple random seeds. The results in Figure 6 in [Rebuttal PDF](https://anonymous.4open.science/r/Anoymuous_ICLR-657E/VLM_function__ICLR_2026_Response_.pdf) demonstrate that our probing method is highly stable across seeds, with Pearson correlations of the heatmaps reaching 1 for all eight functions in Qwen2.5-VL-3B-Instruct.
> >
> > **Q6. How consistent are subquestion function labels across annotators, and how often did GPT-o3 disagree?**
> > **A.**
> > 1. Consistency across human annotators: In our two-stage human verification pipeline, the function label of each subquestion was independently assigned by three expert annotators. In the first stage, approximately 71.84\% of labels were agreed upon by at least two annotators. For subquestions where two out of three annotators disagreed, the labels were reviewed and corrected in the second stage, after which all three annotators reached full agreement.
> >
> > 2. The subanswers were originally generated by GPT-4.1. In the second step, we used GPT-o3 to verify their correctness, and approximately 38.78\% were found to be in disagreement. These disputed subanswers were then further adjudicated and corrected by human annotators.
> >
> > **Q7. How sensitive are head importance maps to (a) top-k token choice, (b) prompt format/order, and (c) random seeds across probes/interventions?**
> >
> > **A.** 1. Sensitivity to the choice of k: We vary $k \in \{1, 3, 5\}$ and compute Pearson correlations of the resulting attention-head heatmaps. Figure 3 in [Rebuttal PDF](https://anonymous.4open.science/r/Anoymuous_ICLR-657E/VLM_function__ICLR_2026_Response_.pdf) shows that the heatmaps remain highly correlated across choices of k, demonstrating that our method is robust to the exact number of selected tokens.
> > This robustness arises because (1) the activation patterns associated with answering a subquestion are reflected across multiple output tokens, and (2) VLM outputs are short, reducing variance from token choice.
> >
> > 2. Prompt format: We also examined the influence of the main question and contextual input (preceding subquestions and their answers). Figure 5 in [Rebuttal PDF](https://anonymous.4open.science/r/Anoymuous_ICLR-657E/VLM_function__ICLR_2026_Response_.pdf) shows that head importance maps vary noticeably across these changes, highlighting the importance of including both the main question and contextual information when identifying cognitive heads.
> >
> > 3. Random seeds: All probing experiments were repeated across 3 random seeds. Figure 6 in [Rebuttal PDF](https://anonymous.4open.science/r/Anoymuous_ICLR-657E/VLM_function__ICLR_2026_Response_.pdf) demonstrates that the identified cognitive heads remain consistent across seeds, indicating robustness to initialization variance.
> >
> > **Q8. Do you think analogous “functional directions” exist in MLP blocks or cross-attention vs self-attention splits, and does combining head+MLP interventions yield larger, more stable gains?**
> >
> > **A.** Our study focuses on attention heads due to their clear interpretability for causal interventions. Extending this analysis to MLPs or different attention splits is an interesting direction for future work.
> > Prior work [1–2] shows MLP layers also store critical knowledge, suggesting analogous “functional directions” likely exist. We expect combining head and MLP interventions could yield larger, more stable effects, as MLPs complement attention by maintaining information dynamically routed through heads. While beyond the current scope, our methodology provides a natural framework to extend such multi-component interventions in future studies.
> >
> > [1] Meng, Kevin, et al. "Locating and editing factual associations in gpt." Advances in neural information processing systems 35 (2022): 17359-17372.
> >
> > [2] Yao, Yunzhi, et al. "Knowledge circuits in pretrained transformers." Advances in Neural Information Processing Systems 37 (2024): 118571-118602.

---

> > > ### Author Response · Authors · 2025-11-26
> > >
> > > Dear Reviewer,
> > >
> > > Thank you very much for your time and effort in reviewing our paper. Based on your constructive feedback, we have made several revisions to the manuscript (highlighted in blue):
> > >
> > > - We included a statistical analysis (t-test) to demonstrate that our findings are statistically significant (lines 322–323; details in Appendix A.10).
> > >
> > > - We added related work on MLLMs with cognitive vision in the Related Works section (lines 498–499).
> > >
> > > - We incorporated additional experiments using activation patching (two methods) in Section 4.2 (lines 352–369).
> > >
> > > - We added sensitivity analyses on the choice of k in top-k token selection, the choice of LLM, and prompt format (line 215), with further details in Appendix A.9.
> > >
> > > - We added the random seed settings for probing in Appendix A.10 (lines 1239–1241).
> > >
> > > We hope these updates address your concerns and improve the clarity of the manuscript. We sincerely appreciate your helpful comments and would welcome any additional suggestions or further discussion.
> > >
> > > Best regards,
> > >
> > > The Authors

---

### Official Review · Reviewer_Ric2 · 2025-10-31

**Soundness:** 2
**Presentation:** 3
**Contribution:** 2
**Rating:** 4
**Confidence:** 4

**Summary:**

This paper aims to explore the internal working mechanisms of Vision-Language Models (VLMs), specifically analyzing whether their Attention Heads play specific roles in multimodal reasoning that are analogous to human cognitive functions.  To achieve this goal, the authors make two main contributions: Construction of the CogVision Dataset: a new interpretability dataset. Proposal of an Analytical Framework: The authors use a probing-based method to identify 'functional heads'—attention heads that are highly correlated with specific cognitive functions.

**Strengths:**

1. The CogVision provides a structured method for analyzing multi-step, multimodal reasoning processes, which surpasses many standard VQA benchmarks. Mapping reasoning steps to specific cognitive functions provides a tool for fine-grained model analysis.
2. The study spans three different VLM families and multiple model scales, making the conclusions about the "universality" and "intrinsic" nature of functional heads more convincing.

**Weaknesses:**

1. **Overextension of the Core Claim ("Reasoning Like Humans")**: The paper's findings merely demonstrate functional specialization within the VLM, which is not equivalent to reasoning in a human-like manner. Any complex system designed to solve multifaceted tasks is likely to evolve modules for handling specific sub-tasks (like visual processing or mathematical computation). The title poses the question, "Do VLMs reason like humans?" but the research findings (the discovery of sparse functional heads) are far from sufficient to answer this. What the authors have found is "functional specialization," not a "human-like reasoning process."

2. **Flaws in the CogVision Dataset:** Data Source is "LLM Cognition," Not "Human Cognition": The dataset's construction relies heavily on GPT-4.1 to generate sub-questions and chains of thought. This means the "cognitive steps" being analyzed are, in fact, the thought processes of another large language model, not those of humans.

3. **Methodological Ambiguities:** The 8 "cognitive functions" defined in the paper are extremely broad (e.g., "Inference" or "Decision-Making"). Can one attention head truly be responsible for all "Inference"? This classification is simplistic, and the probing task may only be capturing superficial statistical correlations related to these coarse labels.

 In Section 3.1, the authors use another LLM (Qwen3-30B) to "select the top-k most important tokens" for extracting head activations. This selection itself is a black-box process, introducing an unnecessary and unanalyzed confounding variable into the experiment. Why is this the best method? How sensitive are the results to the choice of 'k' or the specific LLM used? This makes the source of the probe's input features questionable.

**Questions:**

See above

---

> ### Author Response · Authors · 2025-11-21
>
> Dear Reviewer,
>
> Thank you very much for your time and effort in reviewing our paper. We have thoughtfully addressed each question you raised and hope our responses alleviate your concerns.
>
> **Q1. Overextension of the Core Claim ("Reasoning Like Humans")**
>
> **A.** We thank the reviewer for pointing out the risk of overextending our core claim. We would like to clarify that our work does not claim that VLMs perform human-like reasoning, nor do we equate the observations and analysis on attention heads with a full human reasoning process.
>
> The original title, "Do VLMs reason like humans?", was intentionally posed as an open inquiry rather than a definitive assertion. The subtitle, "Exploring the Functional Roles of Attention Heads", makes clear the research direction and scope. In the paper, we investigate whether VLMs exhibit structural patterns, such as **functional specialization and hierarchical support among cognitive components**. These findings serve as preliminary evidence of analogous organizational principles, not as proof that VLMs replicate human reasoning.
>
> To avoid potential concern, we have revised the title to **"Investigating The Functional Roles of Attention Heads in Vision Language Models: Evidence for Reasoning Modules"**. We also welcome additional suggestions. In the manuscript, we clarify that our observations lie in: **universal, sparse, and intrinsic cognitive heads across architectures; functional importance of cognitive heads; cross-function interaction and hierarchical structure for functional heads correlations.**
>
> Our findings shed light on the structured cognitive architecture embedded in VLMs, and open avenues for cognitive-inspired model design and analysis.
>
> **Q2. Data Source is "LLM Cognition," Not "Human Cognition"**
>
> **A.** The primary goal of our work is to understand the functional roles of attention heads. We would like to clarify that our work does not claim that the reasoning modules or reasoning processes in humans and LLMs are identical. We also provide an explanation of both the design rationale and the quality-control procedures of the CogVision dataset, which are essential for ensuring reliable analysis of attention-head functionality.
>
> 1. In CogVision, sub-questions are generated by GPT-4.1 using specially designed chain-of-thought prompts. Prior studies [1] have demonstrated the effectiveness of using large pre-trained models for dataset construction due to their strong reasoning capabilities and ability to produce high-quality, scalable annotations .
>
> 2. To ensure the subquestions can reflect the human reasoning, we employ a rigorous two-stage human verification pipeline (as described in lines 151–155):
> In the first stage, three expert annotator independently evaluate whether each subquestion is logically structured and consistent with natural human reasoning. QA pairs with incoherent or inconsistent decompositions are filtered out."
>
> 3. Furthermore, we include additional analyses of surface-form variation across cognitive-function groups. As shown in Figures 1 and 2 in [Rebuttal PDF](https://anonymous.4open.science/r/Anoymuous_ICLR-657E/VLM_function__ICLR_2026_Response_.pdf)
> , the eight functions exhibit wide and overlapping distributions in phrasing patterns and token lengths,
> indicating no systematic surface-form differences. These results suggest that the decomposed subquestions do not merely reflect GPT-4.1’s surface patterns, but capture meaningful variation aligned with the intended cognitive functions.
>
> Thus, while GPT-4.1 is used for initial decomposition, human validation and alignment with established cognitive categories ensures that CogVision is suitable for probing VLMs.
>
> [1] Wang, Xinru, et al. "Human-llm collaborative annotation through effective verification of llm labels." Proceedings of the 2024 CHI Conference on Human Factors in Computing Systems. 2024.

---

> ### Author Response · Authors · 2025-11-21
>
> Thanks for your patience. This is the remaining part.
>
> **Q3. The 8 "cognitive functions" defined in the paper are extremely broad (e.g., "Inference" or "Decision-Making"). Can one attention head truly be responsible for all "Inference"?**
>
> **A.** In our work, we identify a subset of attention heads that are highly related to each cognitive function; no single head is sufficient to implement an entire function.
>
> The effectiveness and functional relevance of these identified cognitive heads are demonstrated from multiple perspectives:
>
> 1. **Subsection "FUNCTIONAL CONTRIBUTIONS OF COGNITIVE HEADS"** shows that masking cognitive heads lead to a substantial decline in performance, whereas
> masking an equal number of random heads results in only minor degradation across all VLMs.
>
> 2. **Figure 4 in the paper** further validate their functional roles: masking the relevant functional heads (e.g., language knowledge recall heads for language knowledge recall task) yields a significantly larger performance drop than masking unrelated heads (e.g., vision knowledge recall), confirming their functional specialization.
>
> 3. **Downstream task interventions** provide additional evidence. Negative intervention experiments, as illustrated in Appendix A.10, show that manipulating cognitive heads selectively affects related functions, further supporting their causal and functional roles.
>
> **Q4. Why is "topk" the best method? How sensitive are the results to the choice of 'k' or the specific LLM used? This makes the source of the probe's input features questionable.**
>
> **A.** We prompt Qwen3-30B to select the five most semantically important tokens in the predicted answer. The **examples in Appendix A.8** show that the selected tokens can semantically represent the full output.
>
> Table 8 in Appendix A.8 shows the effectiveness of top-k token compared to first meaning token. In the following table, we present complement results of masking cognitive heads based on different token selection strategies: first is the first token, last is the last token, meaning_first is the first meaning token (excluding formatting), top-k is the top-k most semantically important tokens. We observe that **top-k token masking leads to the most significant performance drop** when masking the cognitive heads, indicating higher precision in identifying different cognitive heads.
>
> Table. Ablation experiment of topK tokens, first meaningful token, first token and last token on Qwen2.5-VL-3B-Instruct, using LLM-judge.
> | Token  | Inter_Head | Vision Low-Level | Vision High-Level | Vision Recall | Language Info | Language Recall | Math | Inference | Decision |
> |-|-|-|-|-|-|-|-|-|-|
> | First | random | 51.61 | 81.82 | 92.42  | 64.81  | 88.57  | 78.05 | 83.54 | 65.63 |
> | First | cognitive  | 45.16  | 56.06  | 86.36  | 48.15  | 74.29  | 78.05 | 51.90 | 68.75  |
> | Last  | random | 90.32 | 89.39 | 95.45  | 61.11 | 38.57 | 85.37  | 84.81  | 75.00  |
> | Last | cognitive  | 67.74  | 89.39 | 67.74  | 0.06  | 68.57  | 82.93 | 43.04 | 65.63 |
> | Meaning_first | random | 90.32   | 86.36   | 84.85 | 77.78 | 81.43  | 65.85 | 70.89 | 68.75 |
> | Meaning_first | cognitive  | 67.74 | 68.18  | 84.85  | 59.26 | 2.85 | 7.31 | 72.15 | 48.44 |
> | TopK  | random | 87.10  | 82.58  | 86.36 | 59.26 | 85.71 | 82.93 | 91.14  | 81.25  |
> | TopK  |cognitive  | **6.45**  | **16.67**| 75.76| 62.96 | 57.14 | **2.43** | **0.00**  | **3.13**  |
>
> **Sensitivity to the choice of k:** We vary $k \in \{1, 3, 5\}$ and compute Pearson correlations of the resulting attention-head heatmaps. Figure 3 in [Rebuttal PDF](https://anonymous.4open.science/r/Anoymuous_ICLR-657E/VLM_function__ICLR_2026_Response_.pdf) shows that the heatmaps remain highly correlated across choices of k, demonstrating that our method is robust to the exact number of selected tokens.
> This robustness arises because (1) the activation patterns associated with answering a subquestion are reflected across multiple output tokens, and (2) VLM outputs are short, reducing variance from token choice.
>
> **Sensitivity to the choice of LLM:** We further experimented with alternative LLMs for token selection. As shown in the following examples ( will add to the Appendix in the revised version), different LLMs consistently select highly similar semantic tokens, reflecting that modern LLMs share strong capability in identifying key semantic units.
>
> Question: What are the notable visual features of the couch and love seat in the image, such as their shape, trim, and upholstery?
>
> Answer: The couch and love seat in the image are patterned after a classic, elegant style with intrica te detailing and a neutral color palette.
>
> LLM top5 token selection:
>
> **Qwen3-30B: classic, elegant, patterned, detailing, neutral**
>
> **GPT5: classic, elegant, patterned, detailing, neutral**
>
> **Llama3.3-70B: classic, elegant, patterned, detailing, color**

---

> ### Comment · Reviewer_Ric2 · 2025-11-25
> **Reply to authors**
>
> Thans for your reply. The response addresses some of my concerns. I choose to keep my score.

---

> ### Author Response · Authors · 2025-11-26
>
> Dear Reviewer,
>
> We are pleased that our previous reply helped clarify some of your concerns. Based on your feedback, we have made several updates to the manuscript (highlighted in blue), including:
>
> - Revised the title to “Investigating the Functional Roles of Attention Heads in Vision Language Models: Evidence for Reasoning Modules.” We have carefully reviewed the entire manuscript and clarified in the Conclusion (Lines 531–534) that our work does not claim VLMs perform full human-like reasoning.
>
> - Added analyses of surface-form variation across cognitive-function groups (Lines 163–165 in Section 2.3), with additional details provided in Appendix A.3.
>
> - Included more examples for negative intervention in Appendix A.12.
>
> - Added sensitivity analyses for the choice of k in top-k token selection and for the choice of LLM (Line 215), with further details in Appendix A.9.
>
> We hope these updates adequately address your concerns and provide greater clarity. If you still have any concerns, please kindly let us know. We would greatly appreciate any additional suggestions you may have and would welcome continued discussion.
>
> Best regards,
>
> The Authors

---

### Official Review · Reviewer_dv3J · 2025-11-04

**Soundness:** 3
**Presentation:** 3
**Contribution:** 3
**Rating:** 6
**Confidence:** 3

**Summary:**

The paper asks whether current VLMs exhibit human-like, functionally organized reasoning and answers this by building an interpretability pipeline centered on attention heads. It introduces CogVision, a multimodal QA dataset that decomposes each question into CoT-style subquestions, each labeled with one of eight perceptual/cognitive functions (e.g. low-/high-level visual reception, language knowledge recall, math reasoning, decision making), creating supervision aligned with a cognitive hierarchy. Using a probing-based method on several VLM families (Intern, Qwen, Gemma), the authors identify “functional heads” whose activations strongly predict these functions and show these heads are sparse, universal across architectures, and layer-structured. Causal interventions—masking vs. amplifying specific heads—demonstrate that these heads are not just correlated but necessary for the corresponding multimodal reasoning steps, and manipulating them transfers to downstream VQA-style tasks. Overall, the work argues that current VLMs contain an emergent, human-analogous hierarchy of attention heads, offering a handle for designing more interpretable and human-aligned models.

**Strengths:**

1. The paper performs a series of well-controlled manipulations (like versus random), augmented (layer-level) probes, and tests on modality sanity to validate the role of attention heads.
2. Introduces CogVision, a multimodal QA dataset that decomposes each question into chain-of-thought (CoT) subquestions richly human-labeled with eight perceptual and cognitive functions—ranging from low-level and high-level visual reception to inference, math reasoning, and decision-making
3. clear organization, formation, and visualizations

**Weaknesses:**

1. Because each “cognitive function” is inferred from GPT-4.1 decompositions of existing benchmarks, the resulting subQAF groups may differ systematically in surface form—such as question phrasing, token length, or modality density—rather than in underlying cognitive process. More analysis of in-group question diversity or ulternation of piepline can clarify if it's confounded by dataset-level artifacts
2. Section 4.3 briefly notes that (for example) 18 % of heads participate in multiple functions and that early-stage functions support later reasoning, yet these statements remain vague and builds on the assumptions that heads can be first identified independently instead of potential activation or not as a result of an interplay of multiple cognitive abilities as the authors also acknowledge the inherent complexity and interdependency.

**Questions:**

See weakness

---

> ### Author Response · Authors · 2025-11-21
>
> Dear Reviewer,
>
> Thank you for your constructive comments and suggestions. Please find our point-by-point responses to your concerns below.
>
> **Q1. Do subQAF groups differ in surface form? More analysis of question diversity.**
>
> **A.** We appreciate the reviewer’s concern. Although the initial subquestions are generated by GPT-4.1 decomposition, we take additional steps to ensure that the resulting subQAF annotations are not artifacts of automated generation. Specifically, three independent human annotators conduct a verification pass to ensure that:
> Each subquestion is logically valid and reflects a coherent step in natural human reasoning;
> The assigned cognitive function accurately corresponds to the intended mental process.
> This human validation substantially reduces the risk that the cognitive-function labels reflect superficial cues.
>
> Furthermore, to address this concern, we include additional analyses of surface-form variation across cognitive-function groups.
> As shown in Figures 1 and 2 in [Rebuttal PDF](https://anonymous.4open.science/r/Anoymuous_ICLR-657E/VLM_function__ICLR_2026_Response_.pdf)
> , the eight functions exhibit wide and overlapping distributions in phrasing patterns and token lengths, indicating no systematic surface-form differences. These results support that the cognitive groups are not determined by trivial lexical or structural artifacts.
> About modality, as described in Section 2.1, some functions (Low-level Visual Reception, High-level Visual Reception, Visual Knowledge Recall) naturally involve vision, while others relate primarily to language. This modality tendency is intrinsic to the underlying cognitive processes rather than an artifact of the pipeline.
>
> **Q2. Section 4.3 briefly notes that (for example) 18 \% of heads participate in multiple functions and that early-stage functions support later reasoning, yet these statements remain vague and builds on the assumptions that heads can be first identified independently instead of potential activation or not as a result of an interplay of multiple cognitive abilities as the authors also acknowledge the inherent complexity and interdependency.**
>
> **A.** Thanks for the suggestions, we have added further illustrations and clarified this point in the revised manuscript.
>
> In our approach, we quantify and rank the accuracy of attention heads for each cognitive function. Heads with high accuracy scores are identified as cognitive heads for that function. **Thus, with our probing-based method, a head that ranks highly for one function may also exhibit non-negligible importance for others, leading to the phenomenon of "Heads Across Functions."** Notably, even if a head ranks in the top 10\% for multiple cognitive functions, our ranking still reveals a primary function for which it is most diagnostic.
>
> Overall, the observed overlap does not contradict the identification of heads for specific functions. Rather, it reflects the inherent interdependencies and shared contributions among cognitive abilities. Our analysis therefore captures both the specialization of individual heads and their potential multi-functionality, providing a more nuanced understanding of the functional organization in VLMs.

---

> ### Author Response · Authors · 2025-11-26
>
> Dear Reviewer,
>
> Thank you once again for your thoughtful and constructive feedback.
> In the revised manuscript, we have made the following changes to address your concerns (highlighted in blue):
>
> - Added analyses of surface-form variation across cognitive-function groups (Lines 163–165 in Section 2.3), with further details provided in Appendix A.3.
>
> - Included additional illustrations and clarifications for “Heads Across Functions” (Lines 418–422).
>
> We hope these updates address your concerns and provide greater clarity. We would greatly appreciate any further suggestions you may have and would welcome continued discussion.
>
> Best regards,
>
> The Authors

---

### Author Response · Authors · 2025-12-02
**Summary of Revisions**

Dear AC and SAC,

We would like to express our sincere gratitude for your handling of our paper, and for the constructive feedback and suggestions provided by the reviewers. During the rebuttal phase, we received follow-up responses from **Reviewer niyk**, whom acknowledged our clarifications and raised the evaluation rating.

- In this paper, we propose **a novel interpretability framework** to systematically analyze the internal mechanisms of VLMs, focusing on the functional roles of attention heads in multimodal reasoning. (an originality and significance highlighted by **Reviewer niyk**.)

- We introduce **CogVision**, a dataset that decomposes complex multimodal questions into step-by-step subquestions designed to simulate human reasoning through a chain-of-thought paradigm, with each subquestion associated with specific receptive or cognitive functions . (recognized as surpassing many standard VQA benchmarks by **Reviewer Ric2**, and further acknowledged as a strength by **Reviewers dv3J and niyk**.)

- We conduct **extensive experiments on three major VLM families** and reveal the existence of **cognitive heads**, analyzing their properties, functional roles, and importance. We uncover cross-function interactions and hierarchical structures, and validate the causal roles of cognitive heads through both negative and positive intervention experiments. (highlighted as strengths by **Reviewers dv3J, Ric2, and idDf**.)

We have carefully addressed all reviewer concerns and revised the manuscript accordingly.
Below we summarize the updates made in the revised manuscript (all changes highlighted in blue):

**Title Revision (for Reviewers Ric2, niyk):**

We updated the title to “**Investigating the Functional Roles of Attention Heads in Vision Language Models: Evidence for Reasoning Modules**.” We also revised the framing throughout the paper to clarify that the properties of cognitive heads and relationships among cognitive heads observed in VLMs do not imply full human-like reasoning.

**CogVision Dataset (for Reviewers dv3J, Ric2):**

Added an analysis of surface-form variation across cognitive-function groups (Section 2.3, Lines 163–165) with additional details in Appendix A.3.

**Heads Across Functions (for Reviewers dv3J, niyk):**

Included additional illustrations and clarifications for “Heads
Across Functions” (Lines 418–422).

**Sensitivity Analyses (for Reviewers Ric2, idDf):**

Added sensitivity analyses on the choice of k in top-k token selection, the choice of LLM, and prompt format (line 215), with
further details in Appendix A.9.

**More Negative Intervention Examples (for Reviewer Ric2):**

Included more examples for negative intervention in Appendix A.12.

**Statistical analysis, Related Work and additional Experiments (for Reviewer idDf):**

Included a statistical analysis (t-test) to demonstrate that our findings are statistically significant (lines 322–323; details in
Appendix A.10), added related work on MLLMs with cognitive vision in the Related Works section (lines 498–499), added additional experiments using activation patching (two methods) in Section 4.2 (lines 352–369), added the random seed settings for probing in Appendix A.10 (lines 1239–1241).

**Correlation Analyses and Clarifications (for Reviewer niyk):**

Added correlation analyses across cognitive functions (Lines 255–257) and across models (Lines 259–260), with detailed
results provided in Figure 17 of Appendix A.10, more explanations about LLM-based judge (lines 308-309), Table 2 (lines 395-396), and clarification of limitations on CogVision (lines 1022-1025).

Thank you very much for your time and thoughtful consideration.

Sincerely,

The Authors

---

### Meta-Review · Area_Chair_uWqP · 2026-01-07

**Summary:**

**Strengths**:

1. Addresses an original and timely question on whether VLMs exhibit functional specialization at the attention-head level (dv3J, Ric2, idDf).

2. Introduces CogVision, a structured multimodal QA dataset with decomposed subquestions enabling fine-grained analysis (dv3J, Ric2).

3. Employs strong controlled experiments (masking, interventions, random and cross-function controls) showing causal relevance beyond correlations (dv3J, idDf).

**Weaknesses**:

1. Central claims are overstated: results demonstrate functional specialization, not human-like reasoning (Ric2, niyk).

2. Dataset construction concerns: Cognitive functions and subquestions are largely generated via LLMs (e.g., GPT-4.1), raising concerns that the benchmark reflects LLM cognition rather than human cognition and may encode surface-level artifacts (dv3J, Ric2).

3. Conceptual and methodological ambiguities: The defined cognitive functions are broad, head-function overlap is insufficiently analyzed, and assumptions of independent head identification underplay interdependencies (dv3J, Ric2, idDf).

**Reviewer Concerns:**

**Addressed**:

- Central claims are overstated: results demonstrate functional specialization, not human-like reasoning (Ric2, niyk).

**Not fully resolved**:

- Dataset construction concerns: Cognitive functions and subquestions are largely generated via LLMs (e.g., GPT-4.1), raising concerns that the benchmark reflects LLM cognition rather than human cognition and may encode surface-level artifacts (dv3J, Ric2).

- Conceptual and methodological ambiguities: The defined cognitive functions are broad, head-function overlap is insufficiently analyzed, and assumptions of independent head identification underplay interdependencies (dv3J, Ric2, idDf).

**Reviewer Scores:**

- Reviewer dv3J: 6 -> 6
- Reviewer Ric2: 4 -> 4
- Reviewer idDf: 4 -> 4
- Reviewer niyk: 2 -> 4

---

### Decision · Program_Chairs · 2026-01-26

Reject